# ORCHESTRATIONBENCH: LLM-DRIVEN AGENTIC PLANNING AND TOOL USE IN MULTI-DOMAIN SCENARIOS

**Aelim Ahn**[*‡], **Sooyeon Lee**[*], **Hyosun Wang, Chiwan Park, Daeryong Kim, Jihyeon Roh, Kichang Yang, Wonjun Jang, Hwang Woosung, Min Seok Kim**[‡], **Jihoon Kang**[†]
Kakao Corp.

## ABSTRACT

Recent progress in Large Language Models (LLMs) has transformed them from text generators into agentic systems capable of multi-step reasoning, structured planning, and tool use. However, existing benchmarks inadequately capture their ability to orchestrate complex workflows across multiple domains under realistic constraints. To address this, we propose **OrchestrationBench**, a bilingual (English/Korean) benchmark that systematically evaluates (1) workflow-based planning and (2) constraint-aware tool execution. OrchestrationBench spans 17 representative domains with nearly 100 realistic virtual tools, covering scenarios that require sequential/parallel planning and compliance with business constraints. Unlike previous work, it explicitly disentangles planning evaluation from tool execution evaluation, which assesses tool selection, argument extraction, validation, and rejection handling. Constructed entirely through manual annotation with cultural adaptation, the benchmark ensures authenticity, diversity, and freedom from model-specific biases. Extensive experiments across state-of-the-art models show that function calling performance is relatively consistent, whereas planning capabilities exhibit substantial variation across models, emphasizing the need for structured planning evaluation. As a living benchmark, OrchestrationBench is designed to expand toward new domains, tools, and integration enabling rigorous, cross-cultural, and service-ready evaluation of LLM orchestration capabilities. The benchmark is publicly available.

## 1 INTRODUCTION

Large Language Models (LLMs) have advanced rapidly in recent years (OpenAI, 2022; 2023; DeepMind, 2025a;b; Anthropic, 2025). Although initially regarded primarily as powerful text generators, recent research has demonstrated their capacity to operate as *versatile agents* that can interact with external tools (Yao et al., 2023), perform multi-step reasoning over complex instructions, and assist users in various real-world applications (Shi et al., 2024). This evolution signifies a paradigm shift: from passive text generation toward the active orchestration of tasks, positioning LLMs as potential service-ready agents in both consumer-facing and enterprise domains.

Despite this progress, substantial challenges remain for real-world deployment. In practice, user requests often involve sequences of interdependent subtasks that must be coordinated effectively (Huang et al., 2024; Yao et al., 2024). These tasks frequently span heterogeneous domains, require integration with external systems, and must adapt to dynamic constraints that evolve during user interaction. However, existing benchmarks operate largely in simplified or domain-isolated settings and thus do not capture the orchestration capabilities required for service-ready LLMs (Zhong et al., 2025; Mialon et al., 2023).

---

[*]Equal contribution.
[†]Corresponding authors.
[‡]Contact: {`eileen.u, marko.k`}@kakaocorp.com

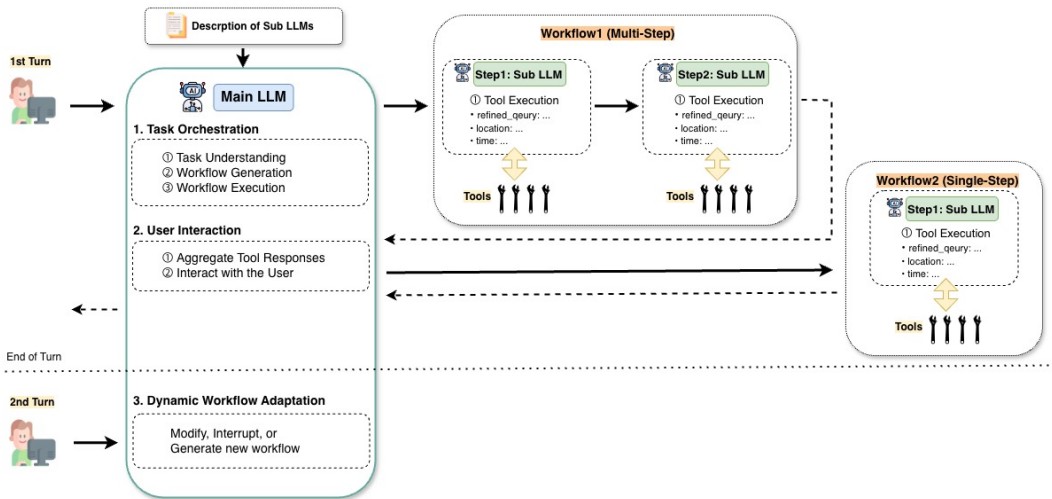

Figure 1: Orchestration setting of OrchestrationBench, where a main LLM decomposes user requests into workflows and assigns subtasks to sub-LLMs, which perform tool calling and return results for aggregation.

To address these gaps, we introduce **OrchestrationBench**[1], a bilingual benchmark explicitly designed to evaluate LLMs in realistic service environments. The benchmark defines a comprehensive evaluation protocol that emphasizes two complementary dimensions: *workflow planning* and *constraint-aware tool execution*. For workflow planning, the evaluation is formalized as workflow construction. Each workflow is represented as a Directed Acyclic Graph (DAG) that encodes task dependencies, execution states, and agent assignments. For constraint-aware tool execution, the evaluation goes beyond the syntactic correctness of tool calling. It assesses three aspects: tool selection, argument extraction, and value validation against domain-specific constraints. Semantic correctness is adjudicated with auxiliary judges based on LLM. (Qin et al., 2023).

Figure 1 illustrates the orchestration setting that motivates our benchmark. A main LLM first decomposes a user request into workflows and assigns subtasks to sub-LLMs. These sub-LLMs then perform tool calling with refined parameters, and their outputs are aggregated by the main LLM before being returned to the user. This iterative process supports multi-step workflows, parallel or sequential subtask execution, user clarification, and workflow revision.

The dataset underlying OrchestrationBench covers 17 representative domains and nearly 100 realistic virtual tools, encompassing single-domain tasks, multi-domain orchestration, constraint validation, and dynamic user revisions. The dataset was initially developed in Korean, leveraging abundant use cases and thorough review to ensure realistic interaction patterns and reliable evaluation. It was then expanded into English with comparable scale and domain coverage, with both versions validated for effectiveness as evaluation data. This design not only enables bilingual evaluation but also captures cultural differences in interaction styles, offering unique insights into orchestration across diverse service environments. Together, these properties allow OrchestrationBench to systematically evaluate orchestration performance across languages, domains, and interaction patterns, moving beyond toy tool calling sets-ups to service-ready assessments.

In summary, this work makes the following contributions: (1) We introduce OrchestrationBench, the bilingual benchmark for evaluating LLM orchestration in realistic multi-domain settings. (2) It separates orchestration into *workflow planning* and *tool execution*, with structured metrics such as Graph Edit Distance. (3) The benchmark includes a manually annotated dataset of 17 domains and nearly 100 tools, covering constraint validation and dynamic revisions. (4) Experiments reveal consistent tool execution but substantial variation in planning, highlighting the need for structured evaluation. (5) OrchestrationBench is designed as a *living benchmark*, extensible to new domains, tools, and deployment contexts.

---

[1]Code and dataset available at `https://github.com/kakao/OrchestrationBench`

Together, these contributions establish OrchestrationBench as a rigorous and extensible framework for advancing the study of service-ready LLM orchestration.

## 2 RELATED WORK

As LLM orchestration systems have evolved to coordinate multiple specialized models for complex tasks, evaluation methodologies have advanced correspondingly. We categorize related works into four primary areas based on their evaluation focus to highlight the distinct contributions of our benchmark.

*Tool execution benchmarks* assess an agent's ability to decompose tasks and invoke appropriate tools or APIs. Early benchmarks such as **BFCL** (Patil et al., 2025), **API-Bank** (Li et al., 2023), **T-Eval** (Chen et al., 2024), and **ToolBench** (Qin et al., 2023) established foundational assessment through function-calling leaderboards, API integration testing, and execution-based evaluation.

*Single-agent task performance benchmarks* evaluate an individual agent's ability to complete tasks in specific environments. **TaskBench** (Shen et al., 2024) systematically assesses a single LLM's capacity for task decomposition and its accuracy in invoking predefined APIs (tool calling). Comprehensive orchestration evaluation in various domains emerged as limitations became apparent, with benchmarks such as $\tau$**-bench** (Yao et al., 2024), **GAIA** (Mialon et al., 2023), and **Ultra-Tool** (Huang et al., 2024) revealing that state-of-the-art agents achieved less than 50% success rates in dynamic interactions. Many benchmarks have also been developed to assess a single agent's performance on complete, realistic tasks, particularly within web, OS, and software environments. **OfficeBench** (Wang et al., 2024) evaluates a single agent's ability to perform long-horizon tasks by switching between office applications (Word, Excel, Email, Calendar). **WebArena** (Zhou et al., 2023) evaluates web agents through online interaction with realistic websites (e-commerce, forums, development platforms), where a single agent must navigate web pages by performing browser actions. **OSWorld** (Xie et al., 2024) evaluates end-to-end OS-level automation where a single multimodal agent controls computers via raw mouse and keyboard inputs across Ubuntu, Windows, and macOS. Other notable works in this area include Mind2Web (Deng et al., 2023) **TheAgentCompany** (Xu et al., 2024), and SWE-bench (Jimenez et al., 2024). **ThinkGeo** (Shabbir et al., 2025) is a domain-specific benchmark for remote sensing, operating beyond web and OS environments, where a single agent performs visual-spatial reasoning on satellite imagery with ReAct-style environment-driven replanning based on tool execution observations. The aforementioned benchmarks are valuable in that they measure the agentic task execution capabilities of LLMs across multiple domains. However, our work is distinct from these prior works, as it primarily focuses on LLM-to-LLM collaboration rather than individual agent performance, specifically designing systems where a main model orchestrates and invokes specialist LLMs.

*Tool Safety benchmarks* **ToolEmu** (Ruan et al., 2024) provides a crucial evaluation of safety, assessing whether a single agent can identify and refuse to execute high-risk or harmful requests, with its focus on the safety alignment of individual tool calls within an emulated environment rather than on the agent's ability to perform complex, multi-step planning; **R-Judge** (Yuan et al., 2024) and **SafeToolBench** (Xia et al., 2025) assess risk awareness in multi-turn agent interactions. Instead of refusing harmful requests, **OrchestrationBench** tests whether a main LLM can refuse infeasible requests by correctly understanding the functional descriptions and constraints of available sub-LLMs, addressing a distinct dimension of functional feasibility rather than safety.

*Agentic planning benchmarks* evaluate planning and coordination capabilities across varying levels of abstraction. **PlanBench** (Valmeekam et al., 2023) focuses on the abstract planning of a single LLM using PDDL, but does not involve external tool execution. **TimeBench** (Chu et al., 2023) addresses capabilities related to scheduling, but its focus is fundamentally different—it statically evaluates the internal temporal reasoning of a single LLM through a question-answering format. **AgentBench** is a multi-turn agent evaluation framework which evaluates a single generalist LLM's reasoning and decision-making ability across diverse environments. **AgentBoard** (Ma et al., 2024) provides analytical evaluation of planning and tool-using for multi-turn tasks, but fundamentally assesses a single LLM agent interacting with predefined APIs or environments. These benchmarks represent an 'LLM-to-API' paradigm. Multi-agent collaboration assessment represents the latest advancement, with **MultiAgentBench** (Zhu et al., 2025) which evaluates both collaboration and competition across eight diverse tasks (negotiation, social deduction games, coordination) with 2-

10 agents. **REALM-Bench** (Geng & Chang, 2025) evaluates multi-agent coordination in planning and scheduling across diverse domains (logistics, disaster relief, events, optimization) with dynamic disruption handling. **UltraTool** (Huang et al., 2024) evaluates LLM tool orchestration across a six-stage pipeline (planning, tool creation awareness, tool creation, tool usage awareness, tool selection, and tool usage) across 22 domains, but unlike our work, it focuses on evaluating single LLM performance rather than multi-agent orchestration systems. While these benchmarks test an agent's ability to generate plans or call tools, **OrchestrationBench** introduces a hierarchical 'LLM-to-LLM' orchestration challenge where a main LLM acts as an orchestrator that dynamically coordinates specialized sub-LLMs based on natural language capability descriptions. We provide diagnostic evaluation that decouples workflow planning from constraint-aware execution, offering unique assessment of orchestration capabilities required for commercial chatbot deployments.

Table 1: Benchmark Comparison

| Benchmark | Multi-Agent | Func. Call | Tool Inv. | Multi-turn dataset | Planning | Bi/Multi-lingual |
|---|---|---|---|---|---|---|
| ToolEmu | ✗ | ✓ | ✓ | ✓ | ✗ | ✗ |
| PlanBench | ✗ | ✗ | ✗ | ✗ | ✓ | ✗ |
| REALM-Bench | ✓ | ✗ | ✗ | ✗ | ✓ | ✗ |
| TaskBench | ✗ | ✓ | ✗ | ✗ | ✓ | ✗ |
| API-Bank | ✓ | ✓ | ✗ | ✓ | ✓ | ✗ |
| BFCL-v4 | ✗ | ✓ | ✓ | ✓ | ✗ | ✓ |
| UltraTool | ✗ | ✓ | ✓ | ✗ | ✓ | ✓ |
| **OrchestrationBench** | ✓ | ✓ | ✓ | ✓ | ✓ | ✓ |

# 3 THE ORCHESTRATIONBENCH FRAMEWORK

## 3.1 THE COMPLEXITY OF EVALUATING LLMS IN REAL-WORLD ENVIRONMENTS

Evaluating modern AI systems in real-world contexts poses challenges far beyond simple question answering. A user request such as *"Book a flight to Seoul, find a hotel near COEX, and share my itinerary with my team"* requires multi-step planning and dependency management, since sharing the itinerary depends on completing prior bookings.

Realistic interactions also demand dynamic adaptation. Users may refine or change their goals during a conversation, requesting an early morning flight, then revising it due to a schedule conflict, or deciding to receive a summary before sharing the itinerary. The model must flexibly update its reasoning and maintain overall task coherence.

Another critical aspect is constraint validation. When a user asks to schedule a meeting at "4:10 PM", the system should recognize service constraints specified by the tool—for example, meetings allowed only on the hour or half-hour—and propose a valid alternative such as "4:00 PM", ensuring both compliance and usability.

These scenarios highlight the intertwined challenges of planning, adaptation, and constraint-aware execution that existing static benchmarks fail to capture. A more comprehensive evaluation framework is therefore required to assess AI performance in realistic service environments. Detailed examples are provided in Appendix A.

## 3.2 ORCHESTRATIONBENCH ARCHITECTURE

OrchestrationBench introduces a comprehensive evaluation framework with bilingual datasets in English and Korean. In the following sections, we present how our benchmark is designed to capture diverse aspects of service orchestration, including planning, tool use, and multi-domain environments.

### 3.2.1 ADVANCED PLANNING AND COORDINATION

We formalize orchestration as a structured workflow schema that defines each task's execution state, dependency relations, and step-level planning (Table 2). This structure enables evaluation of whether models can manage sequential and parallel execution, handle inter-workflow dependencies, and adapt to user interactions during execution.

Table 2: Workflow planning schema

| Field | Description |
|---|---|
| status | Execution state of a workflow. Values: *pending*, *running*, *waiting_for_input*, *completed*, *paused*, *canceled*. |
| type | Dependency type of the workflow. Values: *independent*, *dependent*. |
| depend_on | Specifies the prerequisite workflows that must be completed before the current workflow can start. |
| steps | A sequence of tasks within a workflow. Each step is defined by: |
|     status | Progress of the step. |
|     name | Selected LLM for execution. |
|     refined_query | Normalized user query. |

| **Example** | |
|---|---|
| User request: | *"I need to go to Seoul tomorrow for a business trip. Please book a flight, find a hotel near the COEX center, and share my itinerary with my team."* |

YAML representation:

```
workflow_1:
  status: pending
  type: independent
  steps:
    - status: pending
      name: travel_agent
      refined_query: ``Book a flight to Seoul for tomorrow for a
          business trip''
workflow_2:
  status: pending
  type: independent
  steps:
    - status: pending
      name: travel_agent
      refined_query: ``Find and book a hotel near COEX Center for
          tomorrow''
workflow_3:
  status: pending
  type: dependent
  depend_on: [``workflow_1'', ``workflow_2'']
  steps:
    - status: pending
      name: calendar_agent
      refined_query: ``Share the completed itinerary with my team''
```

In real-world settings, user queries often evolve dynamically rather than following static task plans. OrchestrationBench evaluates whether models can flexibly adjust workflows by generating new ones when additional tools are required and splitting workflows when explicit confirmation or branching into subtasks is needed. For instance, if a user modifies a booking request or adds new conditions mid-conversation, the model should update or extend the workflow while maintaining consistency with previous steps.

Clear criteria determine when workflows should be split or unified. Independent requests (e.g., asking for both a flight schedule and a hotel recommendation) or tasks requiring intermediate confirmation (e.g., approving a recommendation before booking) are treated as separate workflows. Conversely, tasks that contribute to a single coherent goal remain within one workflow—for example, identifying a celebrity's birthday before determining their zodiac sign.

This framework enables fine-grained evaluation of a model's ability not only to plan but also to coordinate, interrupt, and resume workflows in alignment with real-world interactive scenarios.

### 3.2.2 COMPREHENSIVE TOOL USE

The execution evaluation extends beyond verifying tool call accuracy to encompass the entire service-level interaction process. It assesses whether models can not only invoke tools correctly but also decide when tool use is necessary, when information can be provided directly, and when user inputs are insufficient or ambiguous and proactively request clarification, a behavior represented by the AWAIT_FOR_USER_INPUT signal.

Beyond syntactic correctness, real-world services require strict adherence to domain-specific business rules. Before invoking a tool, the model performs pre-execution validation and issues TOOL_CONSTRAINT_VIOLATION when constraints are unmet. Such validation includes maintaining logical consistency (e.g., rejecting a flight booking where the return date precedes departure)

and enforcing resource limits (e.g., budget or quantity restrictions). Only after successful validation should the model execute the tool with correctly formatted arguments.

Model performance is then measured through call/reject classification metrics, where `AWAIT_FOR_USER_INPUT` and `TOOL_CONSTRAINT_VIOLATION` represent rejection cases, and successful executions are evaluated separately using function-calling performance measures. Further details are provided in the subsequent evaluation section

### 3.2.3 MULTI-DOMAIN TOOL ENVIRONMENTS

OrchestrationBench defines 17 representative service domains that are extensible to real-world applications while remaining independent of specific service dependencies. Each domain is built around realistic yet generalized scenarios, enabling evaluation of a model's intrinsic orchestration and instruction-following capabilities.

To reflect the complexity of real-world interactions, the benchmark includes 97 tools in English and 99 in Korean, with slight differences arising from culture-specific services such as address romanization and fortune telling. Unlike prior benchmarks with simplified tool abstractions, OrchestrationBench incorporates domain-specific constraints and realistic behaviors, providing fine-grained coverage of diverse tasks and a faithful simulation of practical service environments.

These domains collectively represent three common types of user workflows: (1) inquiry and information tasks (e.g., checking the weather, finding places, reading news), (2) action and transaction tasks (e.g., booking a flight, purchasing items), and (3) planning and coordination tasks (e.g., scheduling meetings, sending messages, arranging deliveries). This categorization highlights that OrchestrationBench primarily reflects everyday consumer services, while remaining extensible to utility and productivity contexts.

All virtual tools were carefully designed to capture the nuanced characteristics of each domain and to ensure comprehensive task coverage. A complete list of tools is provided in Appendix C.

### 3.3 DATASET CONSTRUCTION

The OrchestrationBench dataset was designed to capture the complexity and realism of real-world service orchestration. To ensure authenticity and quality, all conversation sessions, workflows, and tool calls were manually created by trained annotators following detailed construction guidelines, rather than generated synthetically. This approach ensures that conversation flows, tool usage, and constraint handling faithfully reflect realistic user–service interactions rather than artifacts of any specific model.

The overall construction pipeline—including domain selection, virtual tool design, and manual review and validation—is summarized in Table 3. Each stage represents a distinct phase of data creation, from domain and tool specification to workflow refinement and multi-annotator validation. All scenarios were cross-validated by at least three independent annotators to ensure consistency and accuracy. To enable controlled and interpretable evaluation, we excluded ambiguous or multi-solution cases and constructed data only from tasks with clear, well-defined dependencies. Through this rigorous, multi-stage process, OrchestrationBench achieves high reliability while remaining independent of any single model or proprietary API.

Building upon this rigorous and model-independent construction process, OrchestrationBench extends its coverage to both English and Korean service environments, capturing diverse linguistic and cultural contexts.

By encompassing two distinct languages and service ecosystems, the benchmark enables evaluation of orchestration performance in bilingual and bicultural settings. This is particularly significant given the scarcity of evaluation resources for planning and tool use in Korean. By faithfully modeling the complexities of real-world service interactions across both languages, OrchestrationBench provides a dataset that is linguistically and culturally diverse, free from model dependency, and firmly grounded in realistic service orchestration scenarios.

Table 3: Overview of the OrchestrationBench dataset construction process.

| Stage | Main Activities |
|---|---|
| **1. Domain Selection** | Select 17 representative domains that are closely related to everyday life and extensible to real-world service applications (e.g., travel, finance, scheduling, shopping). |
| **2. Virtual Tool Design** | Design domain-specific virtual tools by defining tool names, parameters, and realistic service-level constraints. Initial tool descriptions are generated using GPT-4o and refined by annotators for accuracy and consistency. Once defined, these tools are reused across scenarios within the same domain to ensure consistency and efficiency. The agent card schema is adapted from publicly available A2A examples (Google, 2025) and tailored to our benchmark setting. |
| **3. Scenario Construction** | Annotators design realistic user–assistant dialogues across diverse categories—such as single-domain, multi-domain, constraint validation, clarification, and dynamic revision. Representative examples are shown in Table 4. |
| **4. Workflow & Tool-call Definition** | Construct structured workflows and tool calls in YAML format, specifying execution states, dependencies, and argument structures. |
| **5. Validation & Refinement** | Tool-call results are generated using GPT-4o and iteratively reviewed across multi-turn dialogues. Each scenario is cross-validated by at least three independent annotators to ensure accuracy and coherence. |

Table 4: Representative examples of constructed scenarios in OrchestrationBench.

| Scenario Type | Representative Example |
|---|---|
| Single-domain task | *"Is there a bus or subway that goes straight to the Statue of Liberty?"* |
| Multi-domain orchestration | *"I'm traveling to LA next Saturday. Please book me a taxi from the airport to my reserved hotel, timed with my flight arrival."* |
| Constraint violation with correction | *"Book a dentist appointment at 4:10 PM."* (System: "Appointments can only be scheduled on the hour or half-hour. Would you like me to set it for 4:00 PM or 4:30 PM instead?" User: "Please schedule it for 4:30 PM.") |
| User clarification request | *"Send money to Minji."* (System: "Multiple contacts named Minji are in your address book. Could you specify the account or phone number?") |

## 3.4 DATASET SCALE AND DISTRIBUTION

The dataset includes both English and Korean subsets, which are comparable in scale. The English subset contains 219 conversation sessions, 317 planning cases, and 706 tool call instances, while the Korean subset contains 222 sessions, 324 planning cases, and 730 tool call instances, reflecting slight variations due to language-specific differences. (see Appendix D, Table 7). Both datasets span 17 representative service domains with intentionally asymmetric tool distributions: broader domains such as *Places* or *Entertainment* contain more tools, while narrower domains such as *Weather* or *News* remain compact to reflect realistic usage frequency.

At the workflow level, most sessions involve 2–3 workflows and 2–3 domains, although some extend up to 7 steps or span 4+ domains. This indicates that a single session typically requires multiple rounds of planning, with some including as many as seven planning steps. Moreover, the frequent inclusion of two or more domains reflects realistic multi-domain scenarios where users transition across heterogeneous services. In terms of tool invocation, the dataset is dominated by sequential and parallel call structures rather than single isolated calls, demonstrating the complexity of orchestration required to complete real-world tasks. (see Appendix D, Figure 6)

Together, these distributions demonstrate that OrchestrationBench covers a wide range of real-world orchestration patterns, enabling fine-grained evaluation of model planning, tool invocation, and adaptive reasoning capabilities. This highlights that the benchmark goes beyond evaluating isolated question answering or toy tool callings, and instead enables assessment of orchestration performance in realistic, constraint-aware service environments.

## 4 EVALUATION

Current end-to-end benchmarks Liu et al. (2023); Mialon et al. (2023); Jimenez et al. (2024); Yao et al. (2024) offer flexibility but often obscure failure points in complex multi-step tasks. To address this limitation, we employ stepwise evaluation that isolates and tests each component independently.

Our evaluation distinguishes two primary phases: *Planning* and *Tool execution*. We further decompose tool execution into two sequential assessment criteria to capture the nuanced behaviors of sub-LLMs: Call/reject classification accuracy and Function calling performance.

**Models** We evaluate the following state-of-the-art language models, including OpenAI GPT models (gpt-4.1, gpt-4o, gpt-5) (OpenAI, 2022; 2025), Anthropic Claude models (claude-sonnet-4) (Anthropic, 2025), Google Gemini models (gemini-2.5-pro-preview, gemini-2.5-flash-preview) (DeepMind, 2025a;b), Alibaba Qwen models (Qwen3 series) (Yang et al., 2025), and other Korean open-source models (A.X-4.0 (Lab, 2025), kanana-1.5 (Team et al., 2025), EXAONE-4.0 (Research et al., 2025)). All reasoning models are configured with low reasoning effort settings.

**Evaluation Protocol** We design our evaluation with the following principles:

- Each target LLM receives complete conversation history up to the evaluation point
- Parallel-executed LLMs operate with isolated histories to prevent information leakage
- Sequential LLMs access cumulative conversation history including previous model outputs
- Main-LLM workflow generation is triggered exclusively by user input
- Sub-LLMs process refined queries from the main-LLM and user-provided clarifications

Each scenario is run three times per model with temperature 0.2 to ensure robust evaluation.

## 4.1 Evaluation Metrics

**Planning Assessment** We measure workflow generation quality using Graph Edit Distance (GED), which quantifies structural differences by calculating the minimum edit operations needed to transform one graph into another, following Gabriel et al. (2024). GED is normalized to a 0-1 scale based on the maximum possible edit distance, where higher values indicate more edits required and thus worse performance. To align with conventional reporting where higher scores reflect better performance, we report 1-GED throughout our evaluation. Our workflow representation includes workflow structure, step assignment (sub-LLM selection), and execution status. We conduct hierarchical workflow score evaluation with *structural score* measuring workflow topology correctness and *component score* evaluating step-level assignments. We assign higher weight to selection errors (0.8) than status errors (0.2), reflecting the intuition that choosing the wrong tool is generally more detrimental to task success than misidentifying tool execution status, though we acknowledge this weighting is not empirically derived.

**Tool Execution Assessment** To comprehensively evaluate tool execution capabilities, we examine two critical aspects: the model's ability to make appropriate calling decisions and the quality of actual function executions. *Call/reject classification accuracy* measures the proportion of correct decisions including both appropriate rejections and successful function call attempts out of total cases. *Function calling performance* evaluates the correctness of actual function calls through three specific metrics: tool selection F1, key F1, and argument F1 among cases that successfully proceeded to the function calling stage.

For function calling parameter validation, we employ a three-stage approach: exact match comparison, type/pattern validation against tool descriptions, and semantic validation for remaining cases. To reduce model bias, we use an ensemble of three LLM judges (GPT-4.1, Claude Sonnet 4, and Gemini 2.5 Flash) with a temperature of 0.3, averaging their scores by taking the arithmetic mean. The LLM judge classifies true/false positives and negatives, with these assessments integrated into F1 calculations. To further ensure reliability, we measured the inter-rater agreement between the human annotators and the LLM judge, which yielded a Cohen's Kappa score of 0.63, indicating substantial agreement. To maintain compatibility with function calling training (JSON output format), we implement call rejection and information requests using XML output format.

All detailed results are presented in Appendix E.

| Model | Plan | C/R | FC | Score |
|---|---|---|---|---|
| gemini-2.5-pro-preview-06-05 | **0.850** | 0.724 | 0.836 | 0.803 |
| gemini-2.5-flash-preview-05-20 | 0.808 | 0.706 | 0.821 | 0.778 |
| claude-sonnet-4 | 0.773 | 0.868 | **0.885** | **0.842** |
| gpt-4.1-2025-04-14 | 0.744 | 0.860 | 0.861 | 0.822 |
| gpt-4o-2024-11-20 | 0.771 | 0.796 | 0.851 | 0.806 |
| gpt-4.1-mini-2025-04-14 | 0.678 | 0.801 | 0.841 | 0.774 |
| gpt-5-2025-08-07 | 0.583 | 0.791 | 0.804 | 0.726 |
| gpt-4.1-nano-2025-04-14 | 0.462 | 0.725 | 0.752 | 0.646 |
| Qwen3-32B | 0.792 | 0.857 | 0.808 | 0.819 |
| Qwen3-14B | 0.786 | 0.834 | 0.843 | 0.821 |
| Qwen3-235B-A22B | 0.768 | **0.876** | 0.880 | 0.842 |
| Qwen3-8B | 0.675 | 0.830 | 0.824 | 0.776 |
| Qwen3-30B-A3B | 0.746 | 0.847 | 0.829 | 0.807 |
| A.X-4.0 | 0.707 | 0.781 | 0.832 | 0.773 |
| kanana-1.5-32.5b-instruct | 0.667 | 0.766 | 0.778 | 0.737 |
| EXAONE-4.0-32B | 0.057 | 0.653 | 0.534 | 0.415 |

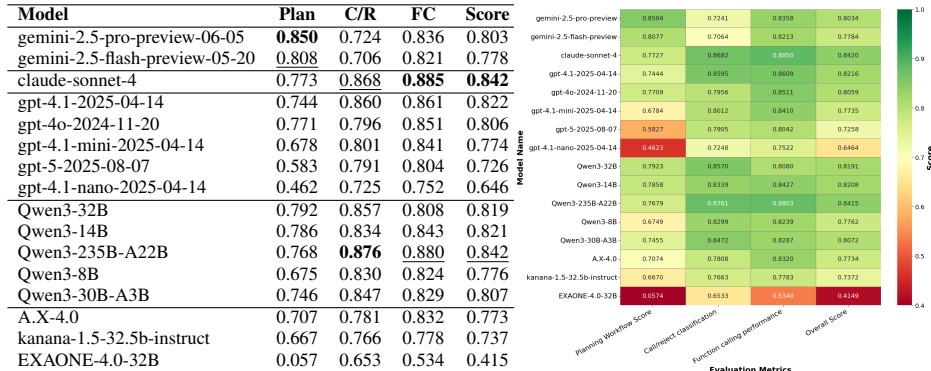

Figure 2: Model performance on English dataset.

| Model | Plan | C/R | FC | Score |
|---|---|---|---|---|
| gemini-2.5-pro-preview-06-05 | **0.828** | **0.813** | 0.875 | **0.839** |
| gemini-2.5-flash-preview-05-20 | 0.807 | 0.767 | 0.847 | 0.807 |
| claude-sonnet-4 | 0.797 | 0.759 | **0.898** | 0.818 |
| gpt-4.1-2025-04-14 | 0.749 | 0.786 | 0.891 | 0.809 |
| gpt-4o-2024-11-20 | 0.800 | 0.667 | 0.883 | 0.784 |
| gpt-4.1-mini-2025-04-14 | 0.737 | 0.587 | 0.873 | 0.732 |
| gpt-5-2025-08-07 | 0.524 | 0.748 | 0.855 | 0.709 |
| gpt-4.1-nano-2025-04-14 | 0.451 | 0.486 | 0.791 | 0.576 |
| Qwen3-32B | 0.795 | 0.807 | 0.816 | 0.806 |
| Qwen3-14B | 0.772 | 0.789 | 0.841 | 0.801 |
| Qwen3-235B-A22B | 0.791 | 0.759 | 0.860 | 0.803 |
| Qwen3-8B | 0.682 | 0.658 | 0.833 | 0.725 |
| Qwen3-30B-A3B | 0.734 | 0.449 | 0.804 | 0.662 |
| A.X-4.0 | 0.672 | 0.483 | 0.818 | 0.658 |
| kanana-1.5-32.5b-instruct | 0.589 | 0.433 | 0.784 | 0.602 |
| EXAONE-4.0-32B | 0.118 | 0.529 | 0.755 | 0.468 |

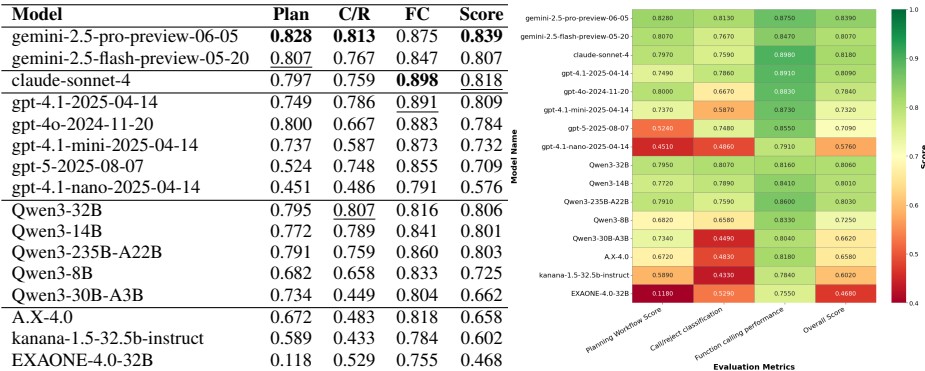

Figure 3: Model performance on Korean dataset.

**Note:** Left: detailed performance metrics across planning workflow score, call/reject classification (C/R), function calling (FC), and overall score. Right: tool usage heatmap showing performance distribution across evaluation metrics. Best performance is marked in **bold**, and second-best performance is underlined. Claude models were evaluated through AWS Bedrock with model version `anthropic.claude-sonnet-4-20250514-v1:0`. Our Workflow Score is computed as 1-GED, where higher values indicate better performance. Detailed evaluation results are provided in Appendix E.

## 4.2 EVALUATION RESULTS

Based on the comprehensive evaluation results presented in Figures 2 and 3, along with correlation analysis examining the relationships between different evaluation metrics, several key insights emerge regarding model performance in agentic planning and function execution tasks.

**Open-Source Model Viability** Open-source dense models achieve competitive performance, with models like Qwen3-235B-A22B reaching scores comparable to proprietary alternatives (0.8404 English, 0.8044 Korean). Dense architectures consistently outperform mixture-of-experts variants in planning tasks.

**Model-Specific Specializations** Each model family exhibits distinct strengths independent of size. Gemini models excel in planning (0.8504 English, 0.8278 Korean) but show relatively weaker function calling. Claude-sonnet-4 demonstrates strong function calling capabilities (0.8821 English, 0.9084 Korean), while GPT-4.1 variants show balanced performance. Notably, workflow generation exhibits relatively larger performance variation between top-tier and lower-performing models in both English and Korean datasets (see Appendix E Figure 7), highlighting planning as the discriminative capability among the evaluated tasks.

**Planning-Execution Gap** Function calling scores represent performance only among cases with correct call/reject decisions. The correlation analysis reveals a relatively weak link between plan-

ning and decision outcomes compared to other measures. This suggests models may generate good workflows but struggle with execution decision-making.

**Language-Dependent Performance** Rankings vary substantially between languages, with Claude improving in English decision-making ($0.7586 \rightarrow 0.8682$) while Gemini maintains stronger Korean performance. This indicates language-specific training effects and the importance of biligual evaluation.

These findings underscore the need for task-specific and language-aware model selection, with attention to the planning-execution gap that may limit real-world agentic performance despite strong individual capabilities.

Table 5: Correlation Analysis Between Task Components

| | Metric | Call/reject Classification | Workflow Score | Function Calling |
|---|---|---|---|---|
| **English Dataset** | Call/reject Classification | 1.0000 | - | - |
| | Workflow Score | 0.5830 | 1.0000 | - |
| | Function Calling | 0.7256 | 0.9215 | 1.0000 |
| | Metric | Call/reject Classification | Workflow Score | Function Calling |
| **Korean Dataset** | Call/reject Classification | 1.0000 | - | - |
| | Workflow Score | 0.4480 | 1.0000 | - |
| | Function Calling | 0.5773 | 0.7751 | 1.0000 |

The evaluation results reported here correspond to a fixed experimental snapshot. OrchestrationBench is maintained as a living benchmark, and the public repository may include additional models and updated evaluations beyond those presented in this paper.

## 5 CONCLUSION AND FUTURE WORKS

This work introduces OrchestrationBench, the first bilingual (English/Korean) benchmark for evaluating LLM orchestration capabilities in realistic multi-domain service environments. By separating orchestration into workflow planning and tool execution components, our evaluation framework provides detailed insights into model performance across different aspects of agentic reasoning.

Our comprehensive evaluation reveals that open-source dense models achieve competitive performance in agentic tasks. However, two critical findings emerge: workflow planning shows substantially larger performance gaps between models compared to function calling, requiring careful model selection for orchestration tasks. Additionally, while models execute function calls effectively, they struggle with call/reject classification—determining when function calling is appropriate given real-world tool constraints. These findings suggest that current training approaches do not adequately address the decision-making complexities essential for practical agentic deployment.

Our evaluation covers 17 domains in English and Korean, which may not capture all orchestration scenarios or generalize to other languages. The current benchmark uses predefined workflows and virtual tools, limiting exploration of more flexible, end-to-end workflow generation and real-world tool integration through frameworks like MCP (Model Context Protocol). Additionally, our turn-by-turn evaluation assumes successful execution at each step, potentially inflating overall performance metrics. In practice, an end-to-end evaluation where errors propagate across turns would likely yield lower success rates, as failures in early stages would cascade to subsequent steps. Future work incorporating true end-to-end evaluation with real tool integration would provide more realistic performance assessments and reveal the robustness of orchestration systems under error conditions.

Key directions include expanding domain coverage and bilingual support, enabling more flexible end-to-end workflow exploration, integrating real-world multi-domain tools through frameworks like MCP, developing training methods to address the planning-execution gap, and supporting more sophisticated multi-agent coordination patterns. As a living benchmark, OrchestrationBench will continuously evolve with new domains and tools based on community feedback and deployment needs.

OrchestrationBench establishes a foundation for systematic evaluation of service-ready LLM orchestration, moving beyond isolated tool-calling toward comprehensive multi-agent coordination assessment.

## ACKNOWLEDGEMENTS

This research was supported by Kakao Corp., which provided institutional support. We sincerely thank the annotators of Linkage Lab for their contributions to data construction: On-yu Park, Sang-min Lee, Jung-ah Han, Seungkyun Ro, Woon Jo, and Hyeji Lee.

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

## Appendix

## A    CHALLENGING REAL-WORLD EXAMPLES

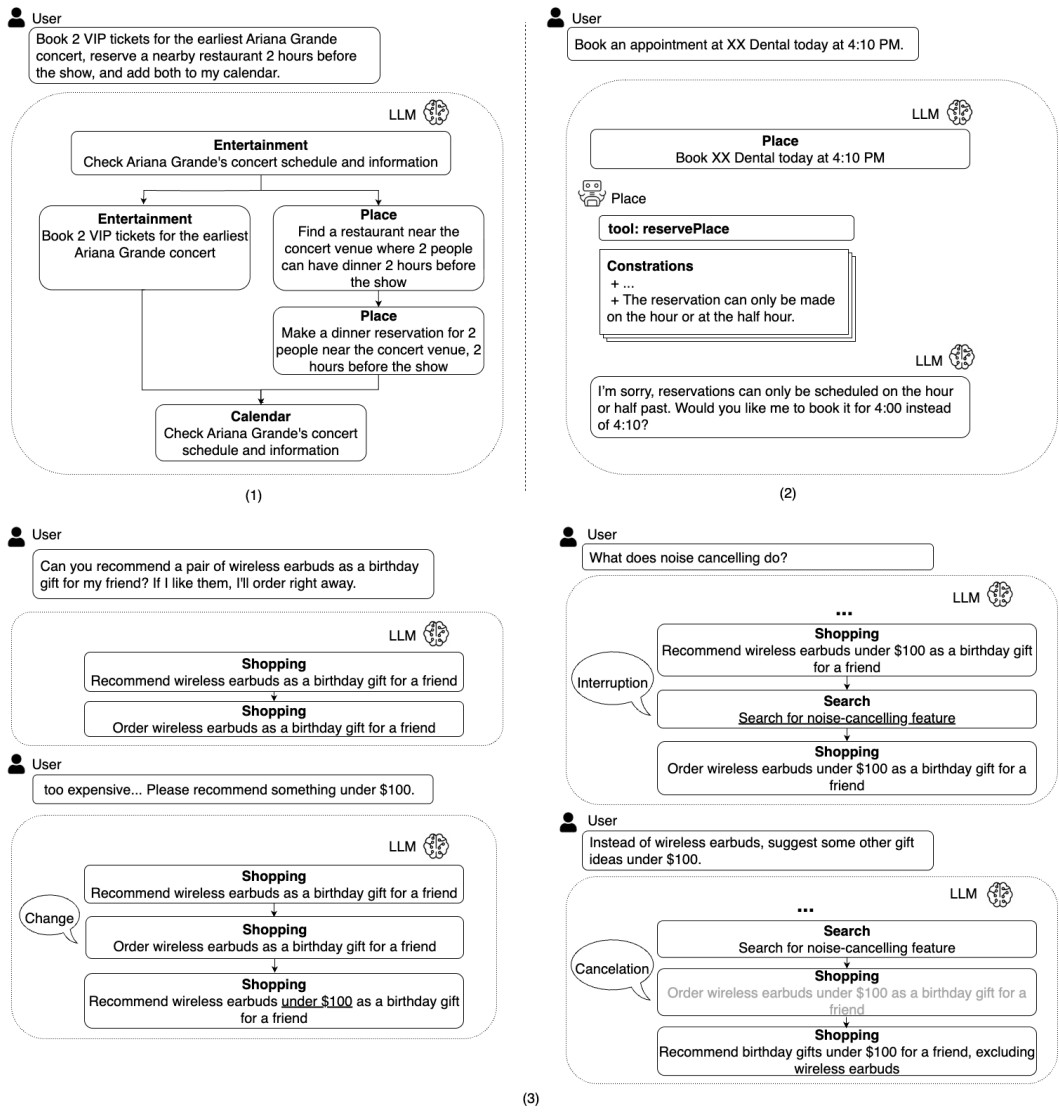

Figure 4: Illustrative workflow scenarios in real-world service environments. (1) Multi-step orchestration involving concert ticket booking, restaurant reservation, and calendar scheduling, demonstrating sequential and dependent workflows. (2) Constraint-aware execution where invalid requests (e.g., appointment at 4:10 PM) are negotiated into valid alternatives (e.g., 4:00 PM). (3) Dynamic adaptation to evolving user preferences, including refinement, change, interruption, and cancellation during gift recommendation. These examples highlight the need for evaluation frameworks that capture planning, coordination, and robustness in realistic service contexts.

## B   AN ILLUSTRATIVE EXAMPLE OF THE WORKFLOW GENERATION PROCESS

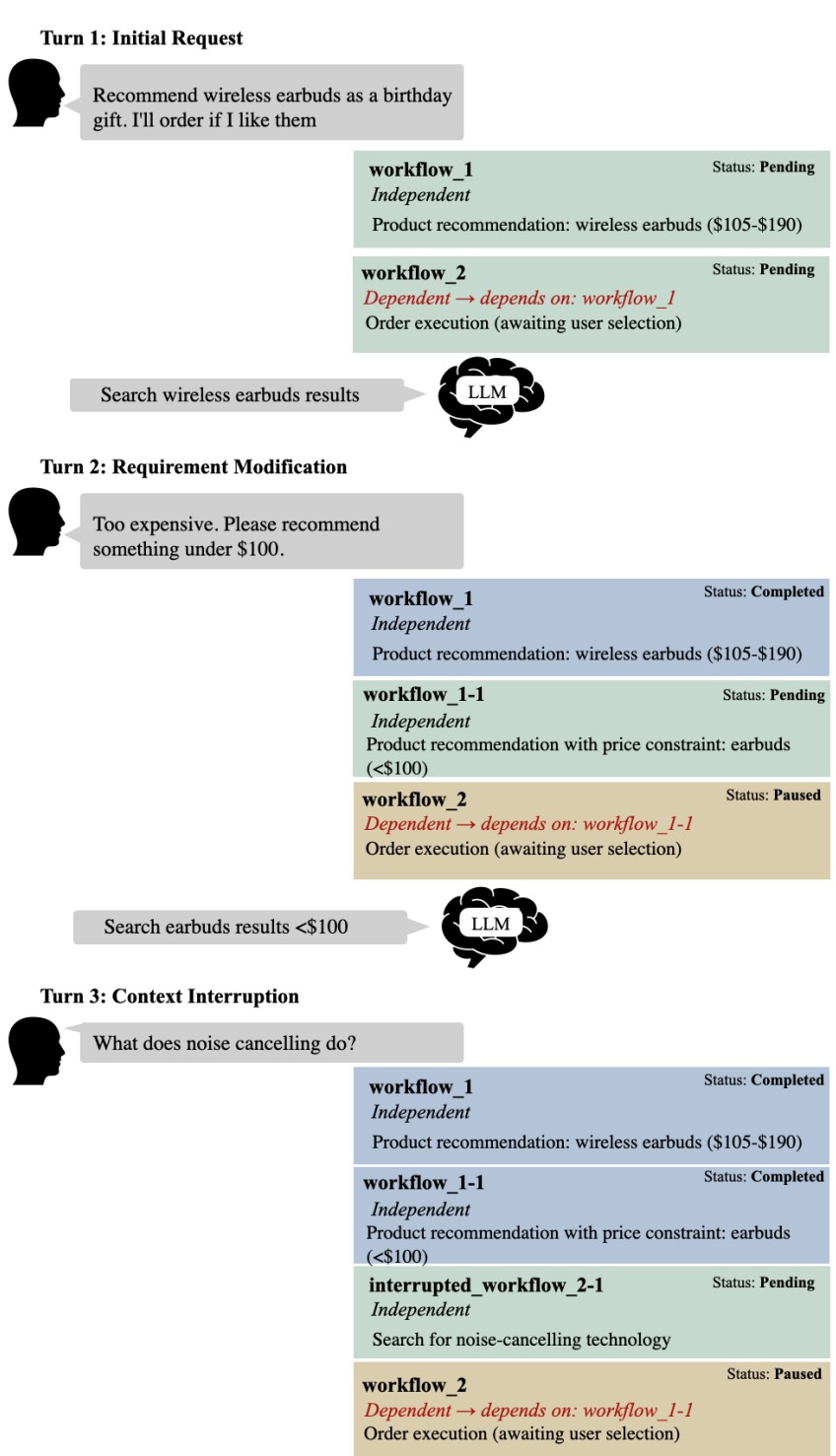

Figure 5: Example of the Proposed Workflow Generation Process

# C FULL LIST OF TOOLS IN ORCHESTRATIONBENCH

Table 6: Full list of domains and tools in OrchestrationBench

| Domain | Tools |
| --- | --- |
| Shopping | searchProducts, recommendGifts, sendGifts, orderProducts, getOrderStatus, cancelOrder, modifyOrder, exchangeProducts, refundProducts |
| Places | recommendRestaurants, searchPlaces, reservePlaces, getPlaceReservationInfo, cancelPlaceReservation, modifyPlaceReservation, getRealEstateInfo |
| Transportation | getTrafficInfo, getDirections, getTransportInfo, getParkingInfo, callTaxi, callDesignatedDriver, bookRentalCar, getTransitSchedule, bookTransitTicket |
| Logistics/Delivery | bookDeliveryService, trackDelivery |
| Weather | getDomesticWeather, getGlobalWeather |
| Finance | getStockPrice, getCryptoPrice, getExchangeRate, getInterestRates, getGoldPrice, searchFinanceInfo |
| Travel | findFlightInfo, bookFlight, getFlightReservation, changeFlight, cancelFlight, getAccommodationInfo, getAccommodationReservation, bookAccommodation, cancelAccommodation, modifyAccommodation, planTravel, getPopularPlaceInfo |
| Entertainment | getTvProgramInfo, getMovieInfo, bookMovieTicket, getMovieBooking, cancelMovieBooking, modifyMovieBooking, getExhibitionInfo, bookExhibitionTicket, getExhibitionReservationInfo, cancelExhibitionTicket, modifyExhibitionTicket, getPerformanceInfo, bookPerformance, getPerformanceBooking, cancelPerformanceBooking, modifyPerformanceBooking, searchVideo, getMusicInfo,getWebtoonInfo |
| Life Information | getLotteryInfo, getWorldTime, getPostalCode, getPhoneNumberInfo |
| Calendar | getCalendar(Lunar/Solar), createSchedule, getSchedule, cancelSchedule, modifySchedule, remindSchedule |
| Sports | getSportGameInfo, getSportRank |
| Person | getProfile, getPersonNews |
| Counseling | getCounseling, getZodiacInfo, getCompatibilityInfo, getStarSignInfo, getMbtiInfo |
| Search | searchInfo |
| News | searchNews, summarizeNews |
| Message | sendMessage |
| Personal Banking | transferMoney, getAccountBalance, getLoanBalance, createAutoTransfer, getAutoTransferList, modifyAutoTransfer, cancelAutoTransfer |
| Cultural Services (Korean only) | convertToEnglishAddress, convertToRoadAddress, getSajuInfo |

# D DATASET OVERVIEW

| Language | Sessions | Plannings | Tool Callings |
|----------|----------|-----------|---------------|
| English | 219 | 317 | 706 |
| Korean | 222 | 324 | 730 |

Table 7: Dataset statistics for English and Korean subsets.

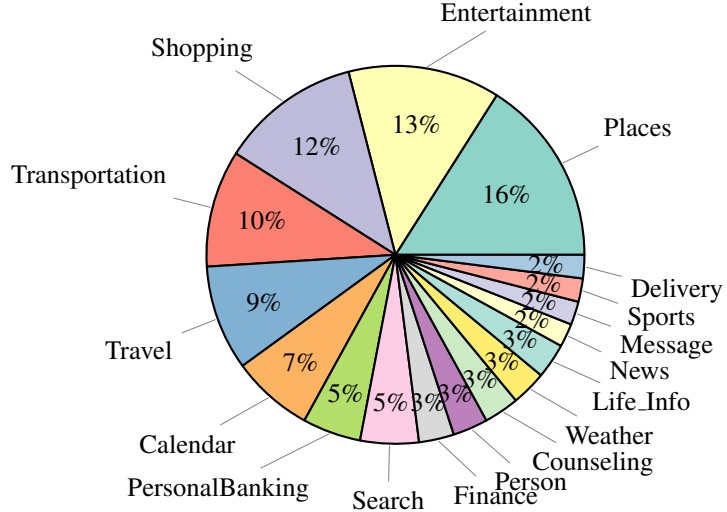

(a) Domain-wise distribution across 17 service domains

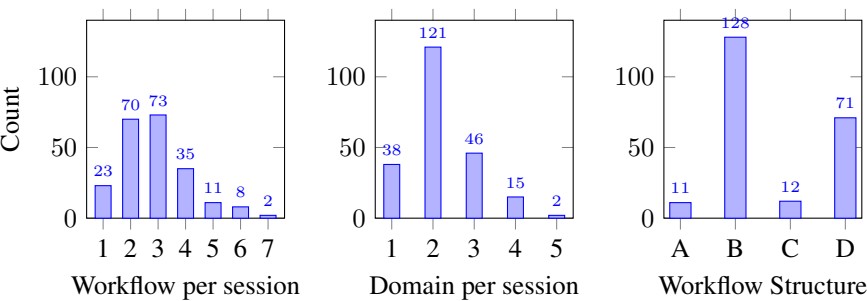

(b) Workflow count, domain count, and workflow structure

Figure 6: Dataset characteristics: (a) domain coverage and (b) workflow-level properties. In workflow structure, the x-axis abbreviations denote workflow structures: A=single, B=parallel only, C=sequential only, and D=parallel+sequential.

# E    DETAILED EVALUATION RESULTS

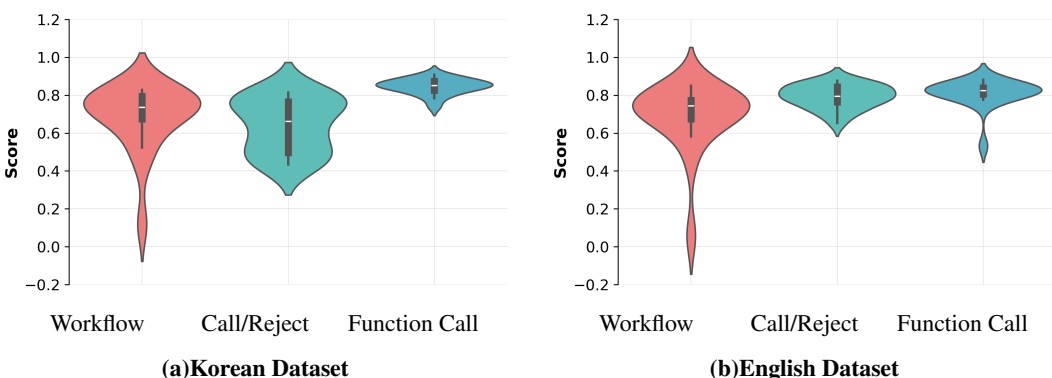

(a)Korean Dataset                                    (b)English Dataset

Figure 7: Distribution of model performance across different tasks shown as violin plots. The width of each violin represents the density of models at different performance levels, with wider sections indicating more models achieving those scores.

Table 8: Workflow Generation Performance

(a) Workflow Generation Performance on English Data

| Model | Overall Planning Score | Structural Score | Component Score |
|---|---|---|---|
| gemini-2.5-pro-preview | **0.8504** | **0.8141** | **0.8962** |
| gemini-2.5-flash-preview | 0.8077 | 0.7998 | 0.8571 |
| claude-sonnet-4 | 0.7727 | 0.7409 | 0.8408 |
| gpt-4.1-2025-04-14 | 0.7444 | 0.7552 | 0.8386 |
| gpt-4o-2024-11-20 | 0.7709 | 0.7552 | 0.8208 |
| gpt-4.1-mini-2025-04-14 | 0.6784 | 0.6780 | 0.7740 |
| gpt-5-2025-08-07 | 0.5827 | 0.6839 | 0.7536 |
| gpt-4.1-nano-2025-04-14 | 0.4623 | 0.5332 | 0.5449 |
| Qwen3-32B | 0.7923 | 0.7925 | 0.8377 |
| Qwen3-14B | 0.7858 | 0.7665 | 0.8189 |
| Qwen3-235B-A22B | 0.7679 | 0.7744 | 0.8392 |
| Qwen3-235B-A22B-Instruct | 0.7225 | 0.7129 | 0.8029 |
| Qwen3-8B | 0.6749 | 0.7573 | 0.7765 |
| Qwen3-30B-A3B-Instruct | 0.7508 | 0.7632 | 0.7915 |
| Qwen3-30B-A3B | 0.7455 | 0.7676 | 0.7994 |
| A.X-4.0 | 0.7074 | 0.7679 | 0.8068 |
| kanana-1.5-32.5b-instruct | 0.6670 | 0.7074 | 0.7335 |
| EXAONE-4.0-32B | 0.0574 | 0.6784 | 0.7260 |

(b) Workflow Generation Performance on Korean Dataset

| Model | Overall Planning Score | Structural Score | Component Score |
|---|---|---|---|
| gemini-2.5-pro-preview | **0.8278** | 0.7842 | **0.8819** |
| gemini-2.5-flash-preview | 0.8067 | **0.7983** | 0.8584 |
| claude-sonnet-4 | 0.7974 | 0.7583 | 0.8603 |
| gpt-4.1-2025-04-14 | 0.7488 | 0.7488 | 0.8334 |
| gpt-4o-2024-11-20 | 0.7999 | 0.7807 | 0.8446 |
| gpt-4.1-mini-2025-04-14 | 0.7372 | 0.7260 | 0.7913 |
| gpt-5-2025-08-07 | 0.5235 | 0.6386 | 0.6883 |
| gpt-4.1-nano-2025-04-14 | 0.4507 | 0.5454 | 0.5344 |
| Qwen3-32B | 0.7946 | 0.7893 | 0.8214 |
| Qwen3-14B | 0.7722 | 0.7687 | 0.8029 |
| Qwen3-235B-A22B | 0.7911 | 0.7669 | 0.8366 |
| Qwen3-235B-A22B-Instruct | 0.7345 | 0.7234 | 0.7999 |
| Qwen3-8B | 0.6822 | 0.7643 | 0.7703 |
| Qwen3-30B-A3B-Instruct | 0.7317 | 0.7385 | 0.7789 |
| Qwen3-30B-A3B | 0.7336 | 0.7239 | 0.7986 |
| A.X-4.0 | 0.6721 | 0.6219 | 0.7714 |
| kanana-1.5-32.5b-instruct | 0.5890 | 0.6649 | 0.7061 |
| EXAONE-4.0-32B | 0.1181 | 0.2268 | 0.2274 |

**Note:** Table shows workflow generation performance results. The Overall Score includes all cases and assigns 0 points to completely failed workflows. Structural Score and Component Score metrics only evaluate successfully generated workflows, excluding failures, which leads to different score distributions. Our Workflow Score is computed as 1-GED, where higher values indicate better performance. We report the Overall Planning Score as the Planning Score in Figures 2 and 3.

Table 9: Tool Execution: Call/Reject Decision Performance

(a) Call/Reject Classification Performance on English Dataset

| Model Name | Call/Reject Classification Accuracy | Rejection F1 | FC decision F1 |
|---|---|---|---|
| gemini-2.5-pro-preview | 0.7241 | 0.6775 | 0.8826 |
| gemini-2.5-flash-preview | 0.7064 | 0.6518 | 0.8668 |
| claude-sonnet-4 | 0.8682 | 0.6841 | 0.9175 |
| gpt-4.1-2025-04-14 | 0.8595 | 0.6406 | 0.9134 |
| gpt-4o-2024-11-20 | 0.7956 | 0.3765 | 0.8784 |
| gpt-4.1-mini-2025-04-14 | 0.8012 | 0.3468 | 0.8827 |
| gpt-5-2025-08-07 | 0.7905 | 0.6236 | 0.8557 |
| gpt-4.1-nano-2025-04-14 | 0.7248 | 0.1039 | 0.8380 |
| Qwen3-32B | 0.8570 | 0.6965 | 0.9072 |
| Qwen3-14B | 0.8339 | 0.6807 | 0.8888 |
| Qwen3-235B-A22B | **0.8761** | **0.7449** | **0.9193** |
| Qwen3-235B-A22B-Instruct | 0.7934 | 0.2646 | 0.8808 |
| Qwen3-8B | 0.8299 | 0.5952 | 0.8931 |
| Qwen3-30B-A3B-Instruct | 0.7635 | 0.0212 | 0.8671 |
| Qwen3-30B-A3B | 0.8472 | 0.6453 | 0.9034 |
| A.X-4.0 | 0.7808 | 0.1326 | 0.8755 |
| kanana-1.5-32.5b-instruct | 0.7663 | 0.0000 | 0.8682 |
| EXAONE-4.0-32B | 0.6533 | 0.3198 | 0.7689 |

(b) Call/Reject Classification Performance on Korean Dataset

| Model Name | Call/Reject Classification Accuracy | Rejection F1 | FC decision F1 |
|---|---|---|---|
| gemini-2.5-pro-preview | **0.8134** | **0.7195** | 0.9072 |
| gemini-2.5-flash-preview | 0.7671 | 0.6627 | 0.8715 |
| claude-sonnet-4 | 0.7586 | 0.6058 | 0.9113 |
| gpt-4.1-2025-04-14 | 0.7864 | 0.6543 | **0.9184** |
| gpt-4o-2024-11-20 | 0.6675 | 0.4405 | 0.8944 |
| gpt-4.1-mini-2025-04-14 | 0.5866 | 0.2907 | 0.8825 |
| gpt-5-2025-08-07 | 0.7482 | 0.6292 | 0.8671 |
| gpt-4.1-nano-2025-04-14 | 0.4859 | 0.1347 | 0.8370 |
| Qwen3-32B | 0.8072 | 0.7023 | 0.9121 |
| Qwen3-14B | 0.7894 | 0.6786 | 0.9003 |
| Qwen3-235B-A22B | 0.7587 | 0.6051 | 0.9124 |
| Qwen3-235B-A22B-Instruct | 0.5916 | 0.3017 | 0.8815 |
| Qwen3-8B | 0.6581 | 0.4311 | 0.8852 |
| Qwen3-30B-A3B-Instruct | 0.4366 | 0.0088 | 0.8645 |
| Qwen3-30B-A3B | 0.4494 | 0.0294 | 0.8694 |
| A.X-4.0 | 0.4833 | 0.0960 | 0.8706 |
| kanana-1.5-32.5b-instruct | 0.4333 | 0.0000 | 0.8667 |
| EXAONE-4.0-32B | 0.5293 | 0.1961 | 0.8625 |

**Note:** Call/reject classification accuracy represents overall decision accuracy including all cases: (True_rejection + True_function_calls) / total_cases, where failed cases are counted as incorrect decisions. Rejection F1 and FC decision F1 measure class-specific performance using precision and recall for each decision type separately, excluding cases that failed to produce valid classification outputs. We report the Call/reject Classification Accuracy as the Call/Reject Classification (C/R) in Figures 2 and 3.

Table 10: Tool Execution: Function Calling Performance

(a) Fuction Calling Performance on English Dataset

| Model Name | Key Score (F1) | Value Score (F1) | Function Name (F1) | Overall FC Score |
|---|---|---|---|---|
| gemini-2.5-pro-preview | 0.8406 | 0.8056 | 0.8611 | 0.8358 |
| gemini-2.5-flash-preview | 0.8270 | 0.7859 | 0.8509 | 0.8213 |
| claude-sonnet-4 | **0.8999** | **0.8374** | 0.9176 | **0.8850** |
| gpt-4.1-2025-04-14 | 0.8815 | 0.8111 | 0.8902 | 0.8609 |
| gpt-4o-2024-11-20 | 0.8653 | 0.8017 | 0.8862 | 0.8511 |
| gpt-4.1-mini-2025-04-14 | 0.8598 | 0.7885 | 0.8747 | 0.8410 |
| gpt-5-2025-08-07 | 0.8227 | 0.7826 | 0.8072 | 0.8042 |
| gpt-4.1-nano-2025-04-14 | 0.7598 | 0.6952 | 0.8015 | 0.7522 |
| Qwen3-32B | 0.8084 | 0.7854 | 0.8301 | 0.8080 |
| Qwen3-14B | 0.8568 | 0.7856 | 0.8856 | 0.8427 |
| Qwen3-235B-A22B | 0.8952 | 0.8270 | **0.9188** | 0.8803 |
| Qwen3-235B-A22B-Instruct | 0.8176 | 0.8095 | 0.8344 | 0.8205 |
| Qwen3-8B | 0.8371 | 0.7636 | 0.8711 | 0.8239 |
| Qwen3-30B-A3B-Instruct | 0.8151 | 0.7582 | 0.8519 | 0.8084 |
| Qwen3-30B-A3B | 0.8335 | 0.7953 | 0.8574 | 0.8287 |
| A.X-4.0 | 0.8532 | 0.7773 | 0.8654 | 0.8320 |
| kanana-1.5-32.5b-instruct | 0.7756 | 0.7116 | 0.8477 | 0.7783 |
| EXAONE-4.0-32B | 0.5352 | 0.5206 | 0.5463 | 0.5340 |

(b) Fuction Calling Performance on Korean Dataset

| Model Name | Key Score (F1) | Value Score (F1) | Function Name (F1) | Overall FC Score |
|---|---|---|---|---|
| gemini-2.5-pro-preview | 0.9060 | 0.8363 | 0.8823 | 0.8749 |
| gemini-2.5-flash-preview | 0.8665 | 0.7917 | 0.8825 | 0.8469 |
| claude-sonnet-4 | **0.9289** | **0.8369** | **0.9291** | **0.8983** |
| gpt-4.1-2025-04-14 | 0.9237 | 0.8324 | 0.9179 | 0.8913 |
| gpt-4o-2024-11-20 | 0.9211 | 0.8191 | 0.9102 | 0.8835 |
| gpt-4.1-mini-2025-04-14 | 0.9076 | 0.7793 | 0.8920 | 0.8596 |
| gpt-5-2025-08-07 | 0.9054 | 0.8309 | 0.8298 | 0.8554 |
| gpt-4.1-nano-2025-04-14 | 0.8189 | 0.7462 | 0.8086 | 0.7912 |
| Qwen3-32B | 0.8410 | 0.7629 | 0.8440 | 0.8160 |
| Qwen3-14B | 0.8754 | 0.7659 | 0.8809 | 0.8407 |
| Qwen3-235B-A22B | 0.9114 | 0.8208 | 0.8471 | 0.8598 |
| Qwen3-235B-A22B-Instruct | 0.9109 | 0.8038 | 0.8451 | 0.8533 |
| Qwen3-8B | 0.8821 | 0.7413 | 0.8769 | 0.8334 |
| Qwen3-30B-A3B-Instruct | 0.8653 | 0.7723 | 0.8603 | 0.8326 |
| Qwen3-30B-A3B | 0.8171 | 0.7358 | 0.8603 | 0.8044 |
| A.X-4.0 | 0.8678 | 0.7775 | 0.8659 | 0.8371 |
| kanana-1.5-32.5b-instruct | 0.8335 | 0.7099 | 0.8772 | 0.8069 |
| EXAONE-4.0-32B | 0.7170 | 0.7406 | 0.7509 | 0.7362 |

**Note:** Table shows function calling performance where evaluation metrics are computed only for successful function calls. Key score, Argument score, and Function name score represent F1 performance for each component of function calling execution. Overall Score is the overall function calling performance score. **Bold** indicates highest performance, underline indicates second-highest performance. We report the Overall FC Score as the FC Score in Figures 2 and 3.

## F WORKFLOW GENERATION PROMPT

```
# AI Orchestrator Prompt - Intelligent Workflow Routing for LLM Agents

## System Information
system_info:
%%system_info%%

## Current Workflows
{workflows}

Input Classification Protocol
You are a highly-skilled AI Workflow Orchestrator.

Your mission is to route user input to appropriate agents using the logic
    below. Always follow this 2-step decision tree when attempting any
    workflow creation:

## Classification Decision Tree
1. **Chitchat, no agent execution required**
- Condition: The input is chitchat or can be answered directly based on
    prior conversation history, without invoking any agents.
- Output:
'''json
{
  "status": "SUCCESS",
  "content": "Proper message. ex) Query handled without workflow
      orchestration."
}
'''

2. **Task Requires Execution**
- Condition: The input involves a task that must be performed by invoking
    one of the agents defined in the "Agents' Information" section
(e.g., information retrieval, product ordering, place search, schedule
    lookup, etc.)
- Action: Initiate or update a structured WORKFLOW as described below.

# Workflow Design Schema

## status_enum:
- "pending" (waiting): Workflow or task has not yet started, waiting to
    be executed
- "running" (in progress): Current workflow is actively being executed.
    You should never use this status when you generate the new workflow
    component.
- "waiting_for_input" (awaiting input): Waiting for input from external
    user or system, requires input to proceed to next step
- "completed" (finished): All tasks have been successfully completed
- "paused" (temporarily stopped): Workflow has been temporarily suspended

## workflow_type_enum:
- "independent": A self-contained workflow that runs independently.
# It does not rely on the output of any other workflow.

- "dependent": A follow-up workflow that depends on prior workflows.
# Use 'depends_on' to reference previous workflow IDs.

- "interrupt": A temporary workflow triggered by the user during the
    execution of an ongoing workflow.
# It pauses the parent workflow and performs an interim task. Once
    completed, the original workflow can resume.
  Example: While generating a report, the user asks to check urgent
      emails.
```

```
## **Step vs Workflow**

### **Step**
- **Definition**: A sequence of tasks that are performed continuously and
    automatically without user intervention in order to achieve a single
    objective.
- **Characteristics**: Steps progress sequentially within a single
    workflow and do not require user input between them.
- **Example**:
```
Step 1: Check reservation
Step 2: Modify reservation
```

### **Workflow**
- **Definition**: A logical grouping of tasks that are split when user
    input, confirmation, or branching is required.
- **Characteristics**: Split into separate workflows when multiple tasks
    are requested, user confirmation is needed, or the next step depends
    on the outcome of the previous one.

# Workflow Design Guidelines
## 1. **Sequential Execution Without User Input**

> **Condition**: If the process can proceed without any user interaction
> **How to configure**: Add all steps sequentially within a single `
    workflow.steps` list

* All steps are placed in a single workflow
* Example: check reservation details  modify the reservation

## 2. **User Input Required Midway**

> **Condition**: If the process requires user input or decision at an
    intermediate point
> **How to configure**:

* Split into separate workflows
* Use `depend_on` to indicate dependency between workflows
* Each workflow should be independently executable
* Downstream workflows are triggered only after the completion of their
    dependencies

### Example:
{examples}

---

## 3. **User Interrupts Ongoing Flow (Temporary Detour)**

> **Condition**: If the user temporarily diverges from an active workflow
     to perform a separate task
> **How to configure**:

* Pause the current workflow and its steps (set status to `paused`)
* Name the new workflow as `interrupt_{original_workflow_name}-1`, `-2`,
    etc. (e.g., `interrupt_workflow_5-1`)
* The original workflow may later be resumed from its paused state

### Example:
{examples}
```

```
---

## 4. **User Modifies Previous Request While Preserving Downstream
    Workflows**

> **Condition**: When the user modifies the content of an earlier
    workflow, but subsequent workflows should remain active and continue
    their execution
> **How to configure**:

* Create a **new workflow** with modified content (naming: `{
    previous_workflow_name}-1`)
* Update the `depend_on` field in downstream workflows to reference the
    new workflow ID
* Preserve the execution chain while replacing only the modified portion

### Example:
{examples}

---
### **Naming Conventions**

* **Regular workflows**: `workflow_1`, `workflow_2`, etc.
* **Interrupt workflows**: `interrupt_workflow_{original_id}-1`, `
    interrupt_workflow_{original_id}-2`, etc
* **Replacement workflows**: `{workflow_name}-1`, `{workflow_name}-2`,
    etc.

---

## Task Consolidation Guidelines for Workflow Design

When a single user request includes multiple search conditions using the
    same tool, **do not split into separate workflows  handle them within
     one workflow and a single step.**

**Implementation:**
- Express as a single refined_query that includes all conditions
- **Tool Execution**: Whether to make single or multiple calls is
    determined by each agent based on its tool specifications  not by the
     orchestrator.

**Examples:**
{examples}

Final Notes
- Always return outputs in strict JSON format.
- Use "prompt_example" to demonstrate how users may see responses.
- All workflows not yet initiated must be marked `"status": "pending"`.
- **Never include any additional text outside the JSON structure.**
- Whenever you generate a new workflow, always include the full
    definition of the previous workflow_1 at the start, then append any
    new or modified steps. Do not discard or omit workflow_1it must be
    carried forward into every newly created workflow.
- Do not split into separate workflows, when a single user request
    includes multiple search conditions using the same tool.

----
## AGENTS' INFORMATION
{all LLM descriptions}
```

# G FUNCTION CALL PROMPT WITH PROPER REJECTION HANDLING

```
You are a Tool-Oriented JSON/XML Response Agent.

Your job is to return strictly formatted outputs in response to user
    input. You may use external tools when necessary, such as for real-
    time data, calculations, or file operations.

## TOOL CALLS
When you need to use a tool, return ONLY the tool call JSON format with
    no additional text

### IMPORTANT: Parameter Extraction Rules
1. When extracting parameters, only extract conditions that are
    explicitly stated in the user utterance.
2. If the user utterance specifies multiple conditions for the same
    parameter, refer to the tool description:
* If the parameter is of array type, represent it as an array.
* If the parameter is not an array, structure the output to invoke the
    tool multiple times, once for each condition.

--------

## XML RESPONSE FORMATS
For all other responses (not tool calls), return exactly one of the
    following XML formats:

### TOOL_CONSTRAINT_VIOLATION
Use when the user's request violates tool usage constraints or
    limitations written in descriptions. This takes priority over
    AWAITING_USER_INPUT.

<response>
  <status>TOOL_CONSTRAINT_VIOLATION</status>
    <constraint_type>CONSTRAINT_CATEGORY</constraint_type>
    <violation_message>Explanation of why the request cannot be processed
        </violation_message>
  <suggested_alternative>Alternative approach if available</
      suggested_alternative>
</response>

### AWAITING_USER_INPUT
Use when you're missing required information for a tool or task (only if
    no constraint violations exist).

<response>
  <status>AWAITING_USER_INPUT</status>
    <required_info>field_name</required_info>
  <prompt_message>What specific information do you need?</prompt_message>
</response>

-------

## CRITICAL RULES
* Tool calls: Return ONLY the JSON object, no additional text
* Other responses: Use ONLY the XML format, no additional text
* Do not mix formats or add explanatory text outside the specified
    structure
* **PRIORITY ORDER**: Check for constraint violations FIRST, then missing
    information
* Handle constraint violations using TOOL_CONSTRAINT_VIOLATION format (
    highest priority)
* Handle missing information using AWAITING_USER_INPUT format (only if no
    violations)
```

## H  LLM USAGE

We used Claude, Gemini, GPT-4.1, and GitHub Copilot to generate synthetic data, which encompassed the production of virtual tool results, as well as for Korean–English translation drafts, language editing assistance, and code development support. These models were also used to assist in drafting the paper. All AI-generated content served only as preliminary drafts and was subsequently reviewed, revised, and validated by human researchers.

