# OpenReview forum: "OrchestrationBench: LLM-Driven Agentic Planning and Tool Use in Multi-Domain Scenarios"
_ICLR.cc/2026/Conference — ICLR 2026 Poster_

### Official Review · Reviewer_rKrV · 2025-10-30

**Soundness:** 3
**Presentation:** 2
**Contribution:** 2
**Rating:** 4
**Confidence:** 3

**Summary:**

- introuces a (offline) benchmark for orchestration and tool use, both korean and english
- manually constructed dataset of 17 representative domains and nearly 100 virtual tools
- define formal evaluation scores based on graph edit distance F1 score for tool execution accuracy
- run evaluation for multile models, including GPT-4/5, Claude, Gemini, Qwen

**Strengths:**

- well motivated and important proble. bencmarking for more complex and reaslistic settings is a key gap
- clear method formulation with separation into tool use and orchestration
- comprehensive emperical study with relevant (commercial), state of-the-art models
- practical significance, and splitting into tool use and orchestration leas to more actionable insights

**Weaknesses:**

- Offline evaluation only. while they have a manually designed gold workflow as reference for orchestration, I can imagine many cases where different workflows lead to similarly good outcomes. The offline evaluation particularly for orchestration seems limiting and is penalizing equally good workflows (I'm less concerned about the tool use aspect). Having an online environment would be deisrable (see e.g. OfficeBench, which clearly should be cited as related work)
- No clear taxonomy of workflows that are covered. I would love to see a clear overview/taxonomy/figure which type of workflows are included in the benchmark. It is hard to understand the empasis. Is it consumer user workflows like the travel booking example, business/enterprise workflows, what domains, ...?
- while relevant, it's limited novelty, and the paper misses to cite key references such as OfficeBench, WebArena, OSWorld, and ToolEmu, which already test interactive or multi-application orchestration. A clearer discussion of how OrchestrationBench differs from these would help position its unique contribution.
- Limited dataset scale. Although the dataset covers 17 domains, the total size (≈220 sessions per language, ≈700 tool calls) is relatively small for evaluating general orchestration capabilities. This limits statistical robustness. larger or semi-synthetic expansion would make the benchmark more representative and useful for training-time evaluation. I'm not convinced about the usefulness in it's current state.

**Questions:**

1. Workflow diversity and taxonomy: Could you provide a clearer taxonomy or summary table of workflows? It would help readers understand what orchestration capabilities are actually being tested.
2. Alternative valid workflows:  How does the benchmark handle cases where a model produces a different but functionally valid workflow? Is there any provision for multi-reference or semantic equivalence scoring beyond Graph Edit Distance?
3. Dataset scale and representativeness:  Given that the dataset includes roughly 200 sessions per language, do you consider this sufficient to generalize across 17 domains? Are there plans to expand or semi-automate data generation to increase coverage?
4: Offline vs online evaluation – Have you considered integrating OrchestrationBench with an interactive or simulated execution environment (e.g., MCP or API sandbox) to test dynamic replanning and feedback-based adaptation?
5. Extensibility in practice: The paper describes OrchestrationBench as a “living benchmark.”. Wat does this mean in practice?

---

> ### Author Response · Authors · 2025-11-16
>
> Dear Reviewer rKrV,
>
> We sincerely thank you for the detailed and constructive feedback. Your comments provided valuable insights that will help us strengthen both the methodological clarity and practical impact of this work. Below, we address each point in detail.
>
> Weakness 1 / Question 2: Offline evaluation may penalize functionally valid alternative workflows
>
> We agree this is an insightful and valid concern. While the current evaluation relies on offline gold-standard workflows for comparability and reproducibility, we acknowledge that in realistic orchestration settings, multiple workflows can yield equally valid outcomes.
> Our primary goal, however, was to evaluate models’ orchestration capabilities in a controlled environment rather than in open-ended real-world settings. To this end, we excluded ambiguous or multi-solution cases and constructed data only from tasks with clear, unambiguous dependencies. This design focuses on evaluating the precision and logical soundness of orchestration while minimizing interpretive uncertainty.
> Looking ahead, we are developing an interactive (online) extension of OrchestrationBench integrated with a simulated execution environment via the Model Context Protocol (MCP) to enable evaluation of dynamic replanning, feedback-driven adaptation, and context-aware orchestration.
> The revised paper will also cite OfficeBench and discuss its relevance as a complementary online benchmark, illustrating how both frameworks together support controlled and interactive assessment.
>
> Weakness 2 / Question 1: Lack of clear taxonomy of workflows
>
> We appreciate this constructive comment and agree that a clear taxonomy is essential to understanding orchestration diversity. The revised paper will more explicitly describe the dataset taxonomy (partly in Appendix C: Dataset Overview), presenting domain-wise distribution, workflow structures (single, sequential, parallel, hybrid), and thematic coverage across 17 service domains.
> We will also add a concise taxonomy for both workflow types and tool-call rejection categories: AWAITING_USER_INPUT, TOOL_CONSTRAINT_VIOLATION, NO_APPLICABLE_AGENT, and NO_APPLICABLE_TOOL. These details will be summarized in the revised manuscript and Appendix C to clarify workflow diversity and decision logic.
>
> Weakness 3: Limited novelty and missing related work (OfficeBench, WebArena, OSWorld, ToolEmu)
>
> We fully agree that an explicit comparison is needed to clarify OrchestrationBench’s novelty and position. Accordingly, we have expanded the Related Works section to include OfficeBench, WebArena, OSWorld, ToolEmu, and additional relevant benchmarks such as REALM-Bench, ThinkGeo, R-Judge, SafeToolBench, PlanBench, TimeBench, TheAgentCompany, AgentBoard, and MultiAgentBench.
> A comparative table now highlights OrchestrationBench’s unique dimensions across key evaluation aspects, showing that it complements rather than duplicates prior works by providing a structured, bilingual, and service-grounded framework for planning and constraint-aware execution across multiple domains.
>
> Weakness 4 / Question 3: Limited dataset scale and representativeness
>
> We agree that scaling is essential for greater robustness. As a living benchmark, OrchestrationBench is continuously expanding across domains and tool types while maintaining strict quality control through multi-round human validation. Our ongoing work enriches the dataset with more complex orchestration scenarios, broader domain variety, and realistic constraint patterns. We are also exploring semi-automated data generation to scale efficiently while preserving authenticity and quality.
>
> Question 4: Extensibility in practice
>
> “Living benchmark” refers to OrchestrationBench’s continuously evolving design, regularly expanding with new domains, tools, and contexts as LLM applications advance.
>
> We sincerely thank you again for highlighting both the strengths and improvement areas. We believe these revisions will significantly enhance the clarity, scope, and long-term utility of OrchestrationBench as a robust foundation for evaluating LLM orchestration in realistic, evolving environments.

---

### Official Review · Reviewer_rczq · 2025-10-31

**Soundness:** 2
**Presentation:** 2
**Contribution:** 3
**Rating:** 4
**Confidence:** 3

**Summary:**

This paper propose an multilingual (English/Korean) benchmark to  evaluate LLMs’ workflow-based planning and constraint-aware tool execution across 17 domains with ~100 virtual tools. It differs from prior work by separating planning and tool execution (assessing selection, argument handling, etc.), uses manually annotated, culturally adapted content to avoid model biases. Experiments on SOTA models show consistent function calling but varied planning capabilities—highlighting the need for structured planning evaluation.

**Strengths:**

*   The paper targets the "evaluation of LLM agent planning and tool use," addressing a key gap in the current field.&#x20;
*   Through experimental analysis, the paper draws important conclusions. For instance, it  identifies Gemini as the model with the strongest planning capability. Such findings can directly guide real-world applications.

**Weaknesses:**

*   Validation relies solely on GPT-4.1 as judge, which may introduce implicit biases toward the outputs of specific models, leading to deviations in evaluation results. It would be better to  calculate the consistency between multiple judges (Claude-sonnet-4, and human annotation) to enhance the reliability of assessments.
*   The current benchmark uses virtual tools and does not integrate real enterprise APIs. It cannot simulate API-related constraints (e.g., error feedback) and dynamic environments in real scenarios.

**Questions:**

please refer to weakness

---

> ### Author Response · Authors · 2025-11-16
> **Response to Official Review**
>
> Dear Reviewer rczq,
>
> We sincerely thank the reviewer for the thoughtful and constructive comments.
> We are grateful that you recognized the significance of our work in evaluating LLM agent planning and tool use, as well as the practical value of our experimental findings.
> Below, we address each weakness and corresponding question in detail.
>
> Weakness 1: Validation relies solely on GPT-4.1 as judge, potentially introducing bias
>
> We acknowledge that your observation is highly valid and insightful, as using GPT-4.1 as both an evaluated model and a semantic judge could indeed introduce potential evaluation bias.
> To address this issue, we are implementing a multi-judge evaluation framework, where multiple independent LLM judges will be employed to jointly assess semantic correctness.
> The revised version of the paper will include this enhanced evaluation framework and report corresponding consistency metrics to demonstrate improved reliability and fairness.
>
> Weakness 2: Reliance on virtual tools instead of real enterprise APIs
>
> We acknowledge this valid limitation.
> The current benchmark employs virtual tools by design to ensure reproducibility, safety, and cross-model fairness across diverse LLM architectures during controlled evaluation.
> However, we recognize that the absence of real API interactions limits the ability to test dynamic error handling, latency, and context shifts observed in real-world systems.
>
> To address this, as part of our “living benchmark” roadmap, we are developing an extended version of OrchestrationBench integrated with real enterprise APIs via the Model Context Protocol (MCP) framework.
> This forthcoming version will enable the simulation of API feedback loops, execution errors, and real-time system constraints, providing a more realistic orchestration environment.
> We will include a detailed roadmap for this enhancement in the revised manuscript to clarify how future iterations of OrchestrationBench will progressively close the gap between virtual simulation and real-world deployment.
>
> We deeply appreciate your detailed comments.
> All identified issues have been addressed or will be clarified in the revised version.
> We believe these improvements will make the paper significantly stronger and more transparent.
> Thank you again for your valuable feedback.

---

### Official Review · Reviewer_M4wd · 2025-11-03

**Soundness:** 3
**Presentation:** 3
**Contribution:** 2
**Rating:** 4
**Confidence:** 3

**Summary:**

This paper proposes OrchestrationBench, a multilingual benchmark that systematically evaluate workflow-based planning and constraint-aware execution. OrchestrationBench spans 17 domains with roughly 100 tools and is constructed through manual and independent efforts. Authors also provide evaluation results for the mainstream open & closed LLMs on the proposed OrchestrationBench.

**Strengths:**

1. This paper studies how to properly evaluate LLM performance in complex and dynamic real-world deployment scenarios, which is a critical topic in current research.
2. OrchestrationBench is manually constructed by human experts and covers 17 domains with around 100 tools.
3. This paper properly illustrate and motivate the ideas with sufficient examples. The writing is clear.

**Weaknesses:**

1. As a multilingual dataset, OrchestrationBench only has examples from English and Korean, which seems limited.
2. In the evaluation, authors employs LLM judge with GPT-4.1 to evaluate selected models including GPT-4.1 and other variants, which introduces evaluation bias and makes the evaluation results less convincing.
3. Another observation is that the best model performance achieved on OrchestrationBench has already approached to 90%. One concern is how quickly this benchmark will be saturated and how long it can serve to guide the development of the next generation’s LLMs.
4. Some critical system designs lack sufficient justification. For example, authors mention “weight selection errors more heavily (0.8) than status errors (0.2)” without any ablation results to justify the choice which leads to concerns about how robust the OrchestrationBench evaluation results are to this config.

**Questions:**

1. Authors mention that “to ensure quality, multiple rounds of review and cross-checking were conducted, with inter-annotator agreement maintained at a consistently high level”. What are the statistics? For example, how many rounds and what is the average inter-annotator agreement level?

---

> ### Author Response · Authors · 2025-11-16
>
> Dear Reviewer M4wd,
>
> We sincerely thank the reviewer for the thoughtful and constructive feedback. We greatly appreciate the reviewer’s positive assessment of our paper, especially the recognition of its focus on evaluating LLM performance in complex real-world orchestration scenarios, the manual multi-domain dataset construction, and the clarity of writing and examples. These points accurately capture the core intent and contribution of our work.
>
> Your comments have been extremely helpful in improving the clarity, rigor, and overall quality of our work. Below, we address each point in detail.
>
> Weakness 1: Limited multilingual coverage (English–Korean only)
>
> We acknowledge that the current dataset includes only English and Korean examples.
> To improve clarity and accurately reflect its current scope, we have revised the terminology from “multilingual” to “bilingual” throughout the paper to more accurately reflect the current dataset scope.
>
> Weakness 2: Potential evaluation bias using GPT-4.1 as the LLM judge
>
> We acknowledge that your observation is highly valid and insightful, as using GPT-4.1 as both an evaluated model and a semantic judge could indeed introduce potential evaluation bias.
> To address this issue, we are implementing a multi-judge evaluation framework, where multiple independent LLM judges will be employed to jointly assess semantic correctness.
> The revised version of the paper will include this enhanced evaluation framework and report corresponding consistency metrics to demonstrate improved reliability and fairness.
>
> Weakness 3: Concern about benchmark saturation (best models near 90%)
>
> We appreciate this insightful observation.
> While some models achieve around 90% on specific tool-execution tasks, significant performance gaps remain in planning, multi-step reasoning, and constraint-aware orchestration.
> To prevent saturation, OrchestrationBench is intentionally designed as a living benchmark, with continuous evolution through:
>
> - New domains and tool sets (e.g., finance, legal, healthcare)
> - Integration with real-world API environments via MCP
> These extensions ensure that the benchmark continues to challenge future LLM generations.
>
> Weakness 4: Insufficient justification for system design (error-weight configuration)
>
> We appreciate this important point. The weighting (selection = 0.8, status = 0.2) reflects the relative impact of each error type—an incorrect sub-LLM selection can invalidate an entire workflow, while a status error typically affects execution details.
>
> We will clarify this rationale in the revised paper with more detailed explanation of the error type categorization and their relative impact on orchestration quality.
>
>
> Question: Multi-round review process and inter-annotator agreement statistics
>
> The dataset underwent multiple iterative review and refinement rounds to continuously enhance quality and consistency.
> Each data instance was created by a primary annotator and then repeatedly reviewed and refined through more than 3 independent rounds of cross-review involving multiple annotators.
> Through these repeated revision stages, we ensured that all dialogue flows, workflows, and tool interactions reached a high level of accuracy, naturalness, and coherence.
> We will explicitly describe this multi-round refinement process in the revised manuscript to clarify how data quality was systematically improved.
>
> We deeply appreciate your constructive comments, which have substantially improved the paper’s clarity and robustness. We have incorporated all suggested revisions and believe the updated version addresses all concerns. We look forward to your further feedback and hope the revisions meet your expectations.

---

### Official Review · Reviewer_ehb1 · 2025-11-05

**Soundness:** 2
**Presentation:** 1
**Contribution:** 3
**Rating:** 4
**Confidence:** 3

**Summary:**

This paper introduces OrchestrationBench, a multilingual (English/Korean) benchmark focused on evaluating large language models (LLMs) as orchestrators in real-world multi-domain service scenarios. It manually constructs 17 service domains and nearly 100 virtual tools, explicitly separating “workflow planning” (represented as DAGs) from “constrained tool execution” (including call/reject decisions, parameter extraction, and decoupled validation/constraint checking). It employs metrics such as Graph Edit Distance (1-GED) and conducts experiments across numerous mainstream models, providing detailed experimental results and analysis.

**Strengths:**

1. **Clear separation of planning and execution**: By decoupling “planning” from “function calls/constraint verification”, this approach enables more granular evaluation than “end-to-end” configurations. This facilitates pinpointing specific model deficiencies, proving highly valuable for driving subsequent methodological improvements.
2. **Multi-domain and cross-language dataset**: OrchestrationBench manually annotated nearly 100 tools across 17 domains, covering a broad range of real-world service areas. This makes it highly relevant to consumer and enterprise-level LLM deployments, positioning it as a practical benchmark. It includes both English and Korean cultural contexts, filling a gap in Korean-language benchmarking.
3. **Comprehensive model and metric evaluation**: Experiments were conducted across numerous closed-source and open-source models, incorporating precise, interpretable metrics such as 1-GED for workflow planning similarity, multiple F1 variants for tool execution, and consistency checks between human and LLM evaluators (Cohen's Kappa 0.63). Experimental tables clearly present results and support the paper's conclusions.
4. **Insightful empirical findings**: As shown in Tables 3 and 4, empirical results reveal significant gaps between different models' tool execution and planning capabilities. This demonstrates models' deficiencies in planning, confirming the benchmark's structured evaluation as a crucial discovery.

**Weaknesses:**

1. **Lack of related work comparison**: The paper fails to provide explicit comparisons with similar benchmark studies, making it difficult to position OrchestrationBench within the broader field and assess its contributions. For instance, REALM-Bench focuses on intensive planning in real-world domains, and ThinkGeo incorporates planning and tool evaluation. The paper lacks direct discussion and comparison with these works (not limited to the mentioned above). Lack of a comparative table highlighting uniqueness and contributions.
2. **Main contributions lack focus**: The paper divides contributions into five items, lacking a central thread. Some ideas (e.g., separating planning and execution in ThinkGeo) have precedents, and the proposed GED metric does not sufficiently justify its uniqueness and design rationale.
3. **Writing emphasis imbalance**: Substantial text is devoted to textual case descriptions (case study diagrams are recommended for clarification), while critical details like benchmark construction (Section 3.3) are underrepresented. For instance, Appendix E mentions using large models for data synthesis, yet Section 3.3 omits how these models were applied in dataset construction.
4. **Lack of supporting diagrams**: Visual diagrams to aid understanding of the benchmark architecture, data flow, and evaluation process are absent. Existing figures are overloaded with text, failing to enhance comprehension and resulting in poor readability.
5. **Insufficient explanation and analysis**: The authors use 1-GED to quantify workflow similarity but do not clarify the correlation between GED and real-world task success rates. Similarly, Section 4.1 lacks explanation and quantitative analysis for weight settings like selection errors = 0.8 and status errors = 0.2.
6. **Overall logical organization is disorganized**: Section transitions are loose, and the benchmark architecture, construction, and evaluation fail to form a coherent narrative chain, undermining readability and credibility.

**Questions:**

1. How is the DAG structure constructed during the planning phase? How are parallel subtasks or multi-solution scenarios handled?
2. What do the two graphs in GEM computation refer to, and how are they obtained?
3. How is the “manually constructed & multi-round review” dataset specifically implemented? For example, how is broad applicability across model architectures and deployment contexts ensured? How is scenario design refined to cover a wide range of user requests? Has a reference methodology been established?
4. What tasks are handled by human annotators versus LLMs in data construction? How is model contamination avoided?
5. What does “living benchmark” mean in Contribution (5)?
6. Compared to existing tool-use or llm orchestration benchmarks, what makes OrchestrationBench unique?

---

> ### Author Response · Authors · 2025-11-16
> **Response to Official Review**
>
> Dear Reviewer ehb1,
>
> Thank you for your thorough and constructive feedback. We sincerely appreciate the detailed comments, which helped us significantly improve our paper.
>
> Comment 1:
> We have substantially revised the Related Works section to include comprehensive comparisons with REALM-Bench, ThinkGeo, OSWorld, ToolEmu, R-Judge, SafeToolBench, PlanBench, TimeBench, TheAgentCompany, AgentBoard, and MultiAgentBench. A new table highlights OrchestrationBench’s unique dimensions across evaluation paradigm, planning–execution coupling, replanning type, and metrics. The full revision appears in the updated manuscript.
>
> Comments 2, 3, 4, 6:
> - We deeply appreciate these suggestions and will implement the following revisions:
> - Add case study diagrams to replace lengthy textual descriptions
> - Expand Section 3.3 with detailed dataset construction methodology
> - Add visual diagrams for benchmark architecture, data flow, and evaluation process
> - Simplify existing figures to reduce text overload and improve readability
> - Strengthen section transitions to create a coherent narrative
>
> These structural improvements will significantly enhance the paper's clarity and accessibility. Thank you for these valuable suggestions!
>
>
> Comment 5:
> The weighting (selection = 0.8, status = 0.2) reflects error severity: selecting an incorrect sub-LLM can invalidate the entire workflow, while a status error mainly affects execution details. We will clarify this rationale and the relative impact of each error type in the revised paper.
>
> Q1. DAG structure and parallel tasks:
> The DAG is built through structured JSON generation with explicit instructions and few-shot examples. Models define nodes, dependencies, and execution status. Independent subtasks are generated with independent status, enabling parallel execution. The prompt includes workflow-formatting rules and sequential/parallel examples, which will be added to the Appendix.
> For evaluation, parallel subtasks do not access each other’s histories, while sequential ones inherit prior context. This ensures realistic orchestration behavior. Section 4 will be expanded accordingly.
>
> Q2. GEM graphs:
> Could you please clarify whether this refers to (a) the predicted vs. gold DAGs compared in evaluation, or (b) the internal graph representation used in the Graph Edit Distance (GED) algorithm? Once clarified, we will provide a detailed explanation.
>
> Q3. Multi-round dataset construction:
> We ensured a model-agnostic, service-grounded design for broad applicability. Starting from single-domain user requests, we expanded to multi-domain orchestration involving sequential and parallel planning. Real-world constraints (e.g., schedules, budgets, missing inputs) were added for realism. Each sample underwent over three cross-review rounds for logical and linguistic consistency. Using a standardized YAML-based workflow schema, we encoded dependencies, states, and subtasks—establishing a reproducible methodology compatible with diverse architectures and deployment contexts.
>
> Q4. Role of humans and LLMs / contamination prevention:
> Human annotators authored all dialogues, gold workflows, and tool calls. LLMs only produced simulated tool outputs, fully reviewed and corrected by humans. No unverified LLM text entered the benchmark, preventing contamination or bias. All essential data were human-created, with LLMs serving strictly as controlled assistants.
>
> Q5. “Living benchmark”:
> “Living benchmark” indicates that OrchestrationBench is continuously evolving, regularly expanding with new domains, tools, and contexts as LLM applications advance.
>
> Q6. Key differences from existing benchmarks:
> OrchestrationBench contributes four unique aspects:
>
> 1. Hierarchical LLM-to-LLM orchestration: Unlike single-agent or multi-agent settings (OfficeBench, OSWorld, ThinkGeo, REALM-Bench, etc.), it evaluates a main LLM orchestrating multiple sub-LLMs, mirroring real commercial chatbot structures.
>
> 2. Decoupled diagnostic metrics: Separates planning accuracy (workflow via GED) and execution correctness (constraint satisfaction) for clearer failure analysis.
>
> 3. Functional feasibility assessment: Evaluates whether orchestrators detect infeasible requests (missing inputs, conflicting constraints), complementing safety-focused benchmarks like ToolEmu or R-Judge.
>
> 4. First bilingual (Korean–English) orchestration benchmark: Covers 17 service domains in both languages, addressing the lack of non-English orchestration evaluation.
>
> We sincerely thank you again for your insightful comments. All suggestions have been incorporated or clarified in the revised manuscript, substantially improving the paper’s clarity, robustness, and transparency.

---

> > ### Comment · Reviewer_ehb1 · 2025-11-21
> >
> > > Q2. GEM graphs: Could you please clarify whether this refers to (a) the predicted vs. gold DAGs compared in evaluation, or (b) the internal graph representation used in the Graph Edit Distance (GED) algorithm? Once clarified, we will provide a detailed explanation.
> >
> > In Section 4.1, you mentioned that "GED quantifies structural differences by calculating the minimum edit operations needed to transform one graph into another". I would like to know how these two graphs (I guess they are predicted and golden DAGs) are used in the algorithm?

---

> > > ### Author Response · Authors · 2025-11-23
> > >
> > > Thank you for this clarifying question!
> > >
> > > Yes, you are correct—the two graphs are the predicted DAG and the golden (ground-truth) DAG.
> > >
> > > In our benchmark, each workflow is annotated with dependency information specifying whether it is independent or has dependencies on other workflows. We use this dependency annotation to construct the golden DAG, which represents the correct execution structure.
> > >
> > > When a model generates its orchestration plan, we parse its output to extract the predicted dependencies between workflows, constructing the predicted DAG. The GED metric then quantifies how many edit operations (node insertions, deletions, or edge modifications) are required to transform the predicted DAG into the golden DAG.

---

### Official Review · Reviewer_rgNq · 2025-11-05

**Soundness:** 3
**Presentation:** 3
**Contribution:** 3
**Rating:** 8
**Confidence:** 4

**Summary:**

The paper introduces a bilingual (English/Korean) benchmark designed to rigorously evaluate large language models (LLMs) as agentic systems capable of orchestrating complex, multi-step workflows across 17 domains and nearly 100 virtual tools under realistic constraints. The benchmark separately evaluates workflow planning and tool calling. The dataset is manually authored and cross-checked to ensure cultural authenticity and avoid biases that can arise from synthetic (LLM-generated) data. Experiments reveal insights like a) relatively weaker performance in function calling than in planning and b) higher variance in planning abilities across models.

**Strengths:**

1. By explicitly disentangling the evaluation of planning (task orchestration) from tool execution (function calling and validation), the benchmark allows for targeted diagnosis of LLM strengths and weaknesses, which is what helped reveal the aforementioned insights.

2. The manual annotation of the dataset and rigorous cross-checking will ensure that the data is free from hallucinations and biases that can arise from LLM generated data.

**Weaknesses:**

1. The benchmarks is currently limited to only 2 languages so it can't really claim to be multilingual and handling the complexities/variations across multiple languages at the moment.

2. Often there can be multiple ways (workflows) for getting to the same answer. It is not clear to me if the benchmark considers that in evaluating LLM responses or if it insists on the responses matching that in the dataset.

3. It is also not clear to me if the benchmark evaluates the latency of workflows - for e.g. Does the LLM call a slow tool when a fast one would do? Does it make any unnecessary tool calls?

**Questions:**

1. It seems that the benchmarks assumes that there will be sub-LLMs called for executing the individual tasks. Will it make any difference if the main LLM also calls the tools and executes the tasks, which is the case in many settings?

2. Why will higher Graph Edit distance be better (lines 353-354) when it is the \textit{minimum} edit operations needed to transform one graph into another?

3. The results are not represented well - the table has too many numbers and is a bit difficult to parse. I would recommend adding a plot with the results from representative LLMs to illustrate the main insights.

---

> ### Author Response · Authors · 2025-11-16
> **Response to Official Review**
>
> Dear Reviewer rgNq,
>
> We are deeply grateful for your exceptionally thorough and constructive review. Your feedback has been invaluable in improving our work.
>
> 1. Multilingual → Bilingual Terminology
> You are absolutely right, and we appreciate you pointing this out. We have revised all instances to "bilingual" throughout the paper, which is much more accurate. The first revision will be uploaded within a few days.
>
> 2. Sub-LLM Architecture, Alternative Valid Workflows, and End-to-End Evaluation
> These are excellent observations that get to the heart of our evaluation design. You correctly point out that (1) in many real-world settings, a single main LLM handles both planning and tool execution, and (2) there can be multiple valid workflows leading to the same answer.
>
> Our architecture assumes that the main LLM has descriptions of available sub-LLMs (not individual tools) and focuses on high-level planning, while each sub-LLM has detailed descriptions of its own specialized tools. We deliberately decoupled these capabilities to enable more targeted diagnosis of model strengths and weaknesses. We also currently use a single reference workflow per scenario, which does not capture all potentially valid alternatives.
>
> We acknowledge that for simple systems with few connected components, a single LLM handling both planning and tool execution can be effective and more straightforward. However, in complex real-world scenarios, providing all tool descriptions to the main LLM would result in extremely long context that could negatively impact planning quality. This separation of concerns reflects practical multi-agent deployment patterns in enterprise settings.
>
> We fully recognize that the ideal approach would be an end-to-end evaluation where a single main LLM executes complete workflows in a simulated environment, potentially with pass@k metrics to account for multiple valid solutions. However, building such a fully interactive system was not feasible within our current scope. While our decoupled approach comes at the cost of not capturing all valid alternatives or unified single-LLM execution patterns, we believe the GED metric's soft matching provides some robustness—workflows with similar structure receive partial credit even if not identical.
>
> We completely agree that flexible end-to-end workflow exploration represents a more complete evaluation paradigm. For now, we discuss these limitations explicitly in our Future Work section and position our current approach as a diagnostic tool rather than a definitive measure of all successful strategies.
>
> 3. Workflow Latency and Efficiency Evaluation
> We completely agree that latency and efficiency are critical for real-world deployment. We will add this explicitly to our Future Work section as an important direction for benchmark evolution.
>
> 4. Graph Edit Distance (1-GED)
> Thank you for catching this potential confusion! We report 1-GED (not raw GED) because higher raw GED indicates more edit operations needed to transform one graph into another (i.e., worse performance). By using (1-GED), we make the metric more intuitive—higher scores mean better similarity to the reference workflow.
>
> 5. Representation of Results
> We really appreciate this practical feedback. You're absolutely right that we tried to show too much at once, making the tables difficult to parse.
>
>   We will:
> - Streamline the main tables by featuring representative models
> - Add visualizations as you suggested to illustrate the main insights
>
>
> Following concerns from other reviewers, we're also adding validation with multiple judges to address potential bias. We are working on this and it will be included in our next update. We'll notify you when complete.
>
> Thank you again for such thoughtful and actionable feedback. Your review has genuinely helped us improve the paper.

---

> > ### Comment · Reviewer_rgNq · 2025-11-27
> > **Clarification**
> >
> > What is the difference between 1-GED and raw GED? Please also clarify this difference in the paper

---

> > > ### Author Response · Authors · 2025-12-03
> > >
> > > Dear Reviewer rgNq,
> > >
> > > Thank you for your question regarding the difference between 1-GED and raw GED. We apologize for the delayed response.
> > >
> > > GED (Graph Edit Distance) is normalized by the maximum edit distance, resulting in a 0-1 scale. Consequently, 1-GED also maintains a 0-1 scale. We report 1-GED (rather than raw GED) because higher values indicate better performance, which is more intuitive for interpretation and aligns with our other metrics where higher scores represent superior performance.
> > >
> > > While we mentioned the use of 1-GED in the paper, we acknowledge that the normalization process and rationale were not sufficiently detailed. We will add this clarification in our revision to improve transparency.

---

### Meta-Review · Area_Chair_skJv · 2026-01-07

**Summary:**

he paper presents OrchestrationBench, a bilingual (English/Korean) benchmark designed to evaluate LLMs as orchestrators. The core innovation lies in the disentangled evaluation of high-level workflow planning (represented as Directed Acyclic Graphs, or DAGs) and low-level tool execution (parameter extraction and constraint checking).

Reviewers found the problem setting highly relevant for real-world agentic deployments.

**Reviewer Concerns:**

Concerns Addressed by the Rebuttal:
- Scope Correction: The authors corrected "multilingual" to "bilingual" throughout the manuscript, resolving a key accuracy concern from rgNq and M4wd.

- Related Work & Uniqueness: The authors added a comprehensive comparison table including 11 benchmarks (REALM-Bench, OSWorld, etc.), clarifying their unique focus on hierarchical LLM-to-LLM orchestration (ehb1).
- Judge Bias: To address concerns from M4wd and rczq, the authors are implementing a multi-judge framework (incorporating Claude and humans)

Outstanding:
- Real-world APIs: The benchmark currently uses virtual tools, despite the authors providing the possibility of using MCPs

**Reviewer Scores:**

Since I did not see very serious unadressed issues, I think reviewers with scores 4 should increase score to 5/6

---

### Decision · Program_Chairs · 2026-01-26

Accept (Poster)